# Global and regional ecological boundaries explain abrupt spatial discontinuities in avian frugivory interactions

**A list of authors and their affiliations appears at the end of the paper**

Species interactions can propagate disturbances across space via direct and indirect effects, potentially connecting species at a global scale. However, ecological and biogeographic boundaries may mitigate this spread by demarcating the limits of ecological networks. We tested whether large-scale ecological boundaries (ecoregions and biomes) and human disturbance gradients increase dissimilarity among plant-frugivore networks, while accounting for background spatial and elevational gradients and differences in network sampling. We assessed network dissimilarity patterns over a broad spatial scale, using 196 quantitative avian frugivory networks (encompassing 1496 plant and 1004 bird species) distributed across 67 ecoregions, 11 biomes, and 6 continents. We show that dissimilarities in species and interaction composition, but not network structure, are greater across ecoregion and biome boundaries and along different levels of human disturbance. Our findings indicate that biogeographic boundaries delineate the world's biodiversity of interactions and likely contribute to mitigating the propagation of disturbances at large spatial scales.

Abiotic gradients underlie the existence of a wide array of natural ecosystems, which are the cornerstone of biological diversity on Earth[1,2]. Ecoregions, defined as regional-scale terrestrial ecosystems[1], delineate regional discontinuities in the environment and in species composition[3,4], whereas biomes mark ecological boundaries at a global scale, such that ecoregions are nested within biomes[1,3] (Supplementary Fig. 1). Accordingly, ecoregion and biome maps have been widely used for guiding conservation planning[3,5], but it has only recently been shown that distinct ecoregions truly represent sharp boundaries for species composition across several taxa[4].

There has been growing recognition that interactions among species are critical for biodiversity and ecosystem functioning[6] and represent an important component of biodiversity themselves, such that interactions may disappear well before the species involved[7]. Species interactions also provide a pathway for the propagation of disturbances via direct and indirect effects, such as secondary extinctions and apparent competition[8,9], with indirect effects of species on others potentially being as important as direct effects[10].

Moreover, adjacent habitats can share many interactions and function as a single dynamic unit[9,11], suggesting that the habitat boundaries typically used by ecologists to delineate interaction networks may not represent true boundaries[11]. Thus, both natural and human disturbances in local communities of interacting species might reverberate and affect ecosystem functioning at multiple sites[12,13], with widespread interactions potentially connecting species at a global scale[12]. However, the spread of disturbances may be hindered when ecological interactions are arranged discontinuously into distinct compartments[14]. Despite this importance, we are only beginning to understand the connections among ecological networks at very large scales[12,13], and it remains unknown whether predictable, large-scale discontinuities in interaction network composition (i.e., the identity of interactions that comprise a local network) exist across ecoregions and biomes. Such discontinuities would mark true network boundaries, and could thus act as a barrier to the global spread of disturbances.

Because species tend to be replaced across ecosystems[2,4] and environmental conditions can favor some types of interactions over

✉e-mail: martinslucas.p@gmail.com; jason.tylianakis@canterbury.ac.nz

others (e.g., by altering the quality and detectability of interaction partners)[15], we hypothesize that ecoregions and biomes delineate the large-scale distribution of species interactions. Specifically, we expect to find sharp differences in the composition of species interactions when crossing ecoregion and biome boundaries, beyond what would be expected from spatial processes alone—which are known to drive gradual changes in species and interaction composition[15]. Indeed, distance−decay relationships have been demonstrated across spatial and elevational gradients not only for species[16], but also for ecological networks[17–19], and likely result from dispersal limitation and increasing environmental dissimilarity with increasing geographic distance[15,16]. Alternatively, ecological boundaries might be blurred by the processes of species and interaction homogenization (i.e., increasing similarity among biological communities), which accompany human disturbances such as land-use change and biotic invasions[12,20]. Thus, an alternative hypothesis would be that shared interactions and biotic homogenization prevent any sharp discontinuities in interaction composition. If this is true, we expect to find a gradual decrease in the similarity of interactions with increasing spatial distance, but no abrupt differences in the identity of interactions from networks located at distinct ecoregions and biomes.

Here we evaluate whether significant changes in the composition of species, the composition of interactions, and the structure of local networks of avian frugivory are explained by large-scale ecological boundaries (ecoregions and biomes) and human disturbance gradients, while accounting for background spatial and elevational effects. Given known patterns of species turnover across environmental gradients[16], we hypothesize a similar pattern of turnover in interaction composition (hereafter, interaction dissimilarity), which could lead to changes in the whole structure of networks (i.e., changes in the arrangement of interactions among species), represented here by a metric combining several descriptors of network architecture, which we call network structural dissimilarity (see "Methods" for more details). Notably, environmental conditions may also affect niche partitioning and interaction specialization, potentially explaining further structural differences among ecological networks from distinct habitats and biogeographical regions[15,21,22]. We focused on avian frugivory networks, that is, local communities of interacting bird and fruiting plant species, because of their importance for seed dispersal[23], promoting species diversity[24] and regenerating degraded ecosystems[25]. As such, mapping the global distribution of plant-frugivore interactions will be crucial to ensure ecosystem functioning and resilience in a context of increasing global changes.

In this study, we show that both ecoregion and biome boundaries explain abrupt spatial discontinuities in the composition of species and their interactions within plant-avian frugivore networks. These effects are detectable on top of the effects of spatial and elevational gradients and after accounting for differences in sampling effort and methods. Similarly, we find evidence that human disturbance gradients also promote large-scale shifts in species and interaction composition. Interestingly, despite the large (often complete) changes observed in the composition of species and interactions, the structure of avian frugivory networks is relatively consistent across large-scale environmental gradients. Our results reveal that ecoregion and biome boundaries delineate the world's biodiversity of interactions and may therefore contribute to mitigating the spread of disturbances across the global network of avian frugivory.

## Results
### Overview of the analysis
To test our hypotheses, we assembled a large-scale database comprising 196 quantitative local networks of avian frugivory (with 9819 links between 1496 plant and 1004 bird species) distributed across 67

ecoregions, 11 biomes, and 6 continents (Supplementary Figs. 1 and 2; Supplementary Table 1). Local networks are composed of nodes−plant and bird species, connected by links whenever two species interact with each other. Each local network is represented by a matrix, with plants and birds on rows and columns, respectively, and cell values describing the weighted network links—the number of fruit-feeding events (i.e., interaction frequency) between a plant and bird species. To ensure that our results would not be driven by taxonomic uncertainty, we standardized the taxonomy of plant and bird species in our local networks. For this, we extracted the frugivore and plant species lists from all networks and performed a series of filters to remove non-existent species names (e.g., morphospecies labels) and standardize synonymous names according to reference databases (steps and examples are presented graphically in Supplementary Figs. 3−6). To account for sampling differences between networks, we controlled statistically for network sampling metrics (i.e., hours, months, years, intensity and methods) in our analyses (see Network sampling dissimilarity section in "Methods"; relationships among sampling variables and network metrics are presented in Supplementary Figs. 7 and 8; variables are described in Supplementary Tables 2 and 3).

We generated several distance matrices ($N \times N$, where $N$ is the number of local networks in our dataset) to be our variables in the statistical models. Specifically, we used ecoregion, biome, local human disturbance (measured using the human footprint index[26]), spatial, elevation and sampling-related distance matrices as predictor variables, and facets of network dissimilarity (i.e., species turnover, interaction dissimilarity, and network structural dissimilarity) as response variables (see a summary of our methods in Fig. 1). By evaluating these three different facets of network dissimilarity, we were able to assess the extent to which changes in species composition are associated with changes in both the identity of component interactions (interaction dissimilarity) and the architecture of local networks (network structural dissimilarity, which may remain the same despite turnover of species and interactions[27,28]). Together these facets contribute to greater understanding of the scale at which one ecological network ends and another begins, and how/why networks vary across large spatial scales[15,27]. We tested the significance of our predictor variables by employing a combination of Generalized Additive Models (GAMs, to allow for non-linear relationships among variables)[29] and Multiple Regression on distance Matrices (MRM, to account for the non-independence associated with pairwise comparisons of local networks)[30]. Essentially, this analysis is equivalent to a GAM, but where the predictor and response variables are distance matrices and the non-independence of distances from each local network is accounted for in the hypothesis testing by permuting the response matrix (see more details in the Statistical analysis section in "Methods"). Finally, we used deviance partitioning analyses to explore the unique and shared contributions of our predictor variables to explaining the variance in network dissimilarity. We did this by fitting reduced models (i.e., GAMs where one or more predictor variables of interest were removed) and comparing the explained deviance.

### Species turnover across networks
Using a binary approach−in which two ecological networks located within the same ecoregion/biome were given a value of zero, otherwise a one−to generate our ecoregion and biome distance matrices, we found that the turnover of plant and frugivorous-bird species composition was strongly affected by ecoregion ($t = -38.093$; $P = 0.001$) and biome ($t = -8.799$; $P = 0.001$) boundaries (Supplementary Table 4). Trends were qualitatively similar when we assessed the effect of these ecological boundaries using a quantitative approach based on the environmental dissimilarity between ecoregions and biomes (Supplementary Table 5; Supplementary Figs. 9a-b). Similarly, there was an overall trend of networks located at different positions along the human disturbance gradient having different species composition

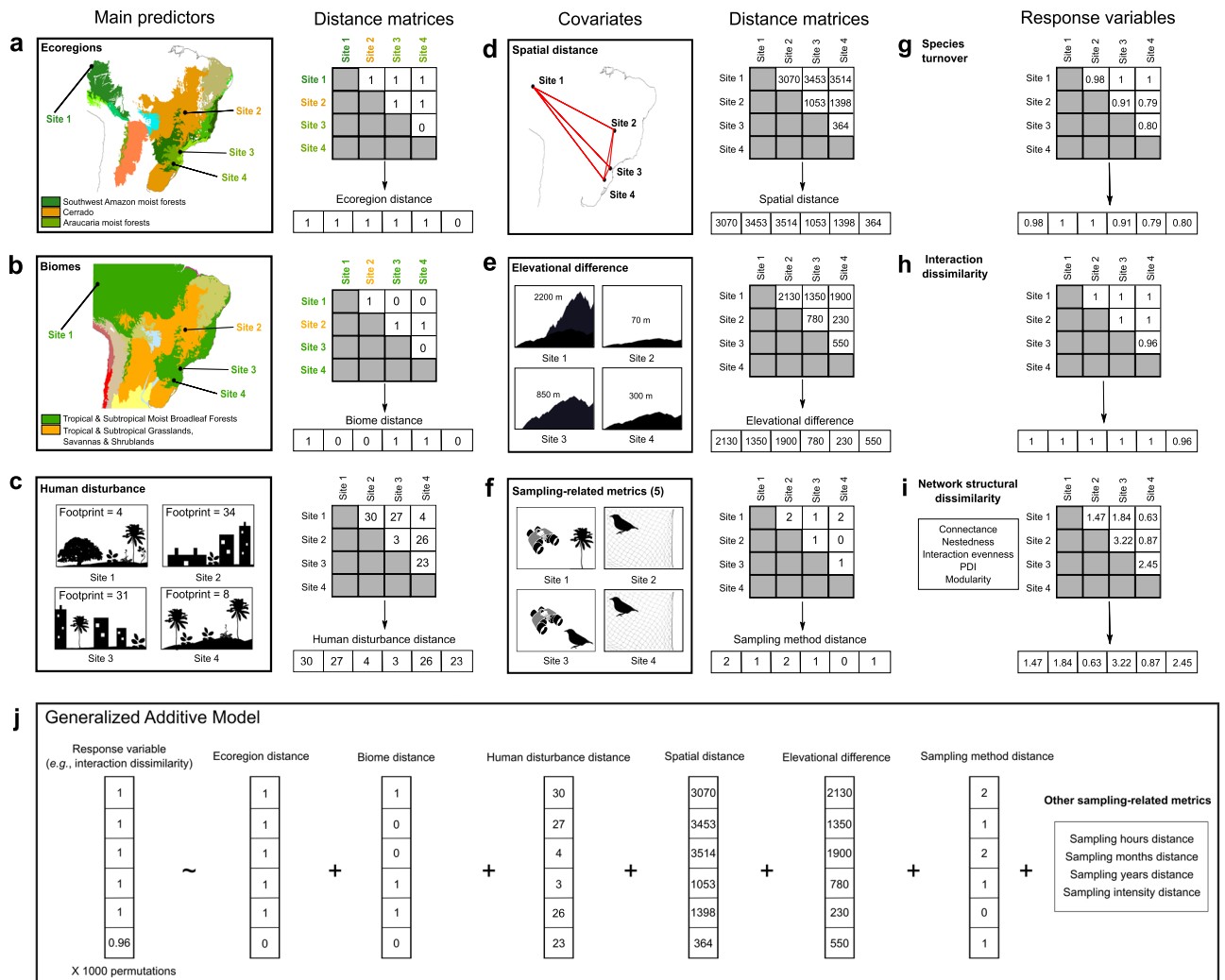

**Fig. 1 | Our approach for evaluating the multiple predictors of network dissimilarity at large spatial scales.** We used several distance matrices ($N \times N$, where $N$ is the number of local networks in our dataset) as variables in the statistical models. **a, b** Maps show examples of ecoregions and biomes (colors of shaded areas) represented in our dataset. Points indicate the locations of four network sites used to illustrate how we generated our distance matrices (see Fig. 2 to visualize the locations of all network sites in our dataset). Ecoregion and biome distance matrices were generated using both a binary (shown in the figure) and a quantitative approach (generated by measuring the environmental dissimilarity between ecoregions/biomes; see "Methods"). Because ecoregions are nested within biomes, network sites located within the same ecoregion are always within the same biome, but the opposite is not necessarily true; see, for example, the comparison between network site 1 and network site 3, which involves two ecoregions (Southwest Amazon moist forest and Araucaria moist forest) from the same biome (Tropical & Subtropical Moist Broadleaf Forests). **c** The human disturbance distance matrix was generated by calculating the absolute difference between local-scale human footprint values around each network site. **d–f** Spatial distance, elevational difference and sampling-related distance metrics (i.e., sampling methods, hours, months, years, and intensity) were used as covariates in our models to control for distance-decay effects and differences in network sampling. Note that even though we only depict the sampling method distance matrix in **f**, all sampling-related metrics were used as predictors in the models. **g–i** We used three different facets of network dissimilarity (i.e., species turnover, interaction dissimilarity and network structural dissimilarity) as response variables (see Network dissimilarity section in "Methods"). **j** We tested the significance of our predictor variables by employing a combination of Generalized Additive Models (GAM) and Multiple Regression on distance Matrices (MRM). In this analysis, the non-independence of distances from each local network is accounted for by performing 1000 permutations of the response matrix. Ecoregions and biomes were defined based on the map developed by Dinerstein et al.[3] (available at https://ecoregions.appspot.com/ under a CC-BY 4.0 license). Bird and plant silhouettes were obtained from http://phylopic.org under a Public Domain license.

($F = 28.504$; $P = 0.001$) (Supplementary Fig. 9c). As expected, spatial and elevational gradients also promoted species turnover across networks (Supplementary Tables 4 and 5), with spatial distance alone accounting for the greatest proportion of deviance explained in species turnover (12.9%), followed by the shared contribution of spatial distance and ecoregion boundaries (11.2%) (Supplementary Fig. 10).

### Interaction dissimilarity

Plant-frugivore interaction dissimilarity increased significantly across ecoregions ($t = -36.401$; $P = 0.001$), biomes ($t = -3.323$; $P = 0.044$) and different levels of human disturbance ($F = 29.988$; $P = 0.001$), even after accounting for the effects of spatial distance, elevational differences, and sampling-related metrics (Table 1). Similar results were found when we performed the analyses using quantitative versions of ecoregion and biome distance matrices (Supplementary Table 6). These findings provide strong support to the hypothesis that large-scale ecological boundaries mark spatially abrupt discontinuities in plant-frugivore interactions (Figs. 2 and 3; Supplementary Fig. 11). Importantly, a great proportion of the deviance explained by biomes was shared with ecoregions (see the overlapping areas between ecoregions and biomes in Fig. 4 and Supplementary Fig. 12), which suggests that changes in interaction dissimilarity across biome boundaries

**Table 1 | Multiple predictors of plant-frugivore interaction dissimilarity ($\beta_{WN}$)**

| Parametric coefficients | Estimate | *t* | *P* |
|---|---|---|---|
| Intercept | 0.997 | 2964.191 | **0.001** |
| Ecoregion (same) | −0.070 | −36.401 | **0.001** |
| Biome (same) | −0.002 | −3.323 | **0.044** |
| **Smooth Terms** | **EDF** | **F** | **P** |
| s (human disturbance distance) | 8.534 | 29.988 | **0.001** |
| s (spatial distance) | 8.785 | 65.378 | **0.001** |
| s (elevational difference) | 6.168 | 47.707 | **0.001** |
| s (hours distance) | 1.558 | 5.449 | 0.290 |
| s (months distance) | 5.482 | 6.902 | 0.075 |
| s (years distance) | 7.208 | 11.848 | **0.019** |
| s (sampling intensity distance) | 1.018 | 5.182 | 0.259 |
| s (methods distance) | 8.632 | 16.002 | **0.005** |

Here, we used the binary version of ecoregion and biome distance matrices. *P* values were calculated using a two-tailed statistical test that combines Generalized Additive Models (GAM) and Multiple Regression on distance Matrices (MRM). In this approach, the non-independence of distances from each local network is accounted for in the hypothesis testing by performing 1000 permutations of the response matrix (see "Methods"). EDF represents the effective degrees of freedom for each smooth term in the model. *N* pairs of networks = 19,110.
Bold *P* values indicate statistically significant results (*P* < 0.05).

mostly reflect the variation occurring at a finer (ecoregion) scale. Specifically, crossing an ecoregion boundary induced an average 7% increase in interaction dissimilarity, while crossing a biome boundary only induced an additional 0.2% change. As with species turnover, we found a strong effect of human disturbance gradients on interaction composition ($F = 29.998$; $P = 0.001$), such that networks at opposite ends of the human disturbance continuum usually exhibited very different interactions, even if they were located within the same ecoregion or biome (Fig. 5; Supplementary Fig. 13).

In addition to the importance of ecological boundaries and human disturbance gradients for driving plant-frugivore interaction dissimilarity, these effects were observed against a background of increasing interaction dissimilarity through space. Indeed, interaction dissimilarity increased sharply until a threshold distance of around 2500 km between network sites, beyond which few networks shared any interactions and dissimilarity remained close to its peak (Fig. 6; Supplementary Fig. 14). In the cases where spatially distant networks shared interactions, these typically involved species that had been introduced in at least one location. For instance, the interaction between the Blackbird *Turdus merula* and the Blackberry *Rubus fruticosus* was shared between networks located more than 18,000 km apart: while both species are native in Europe, they have been introduced by humans to Aotearoa New Zealand. Similarly, networks from Asia were connected to Hawai'i mostly through interactions involving introduced species in the latter, such as the Red-whiskered Bulbul *Pycnonotus jocosus* and the Java Plum *Syzygium cumini* (Fig. 2).

Deviance partitioning revealed that the shared effect of crossing ecoregion boundaries and spatial distance explained the greatest proportion of the variance in plant-frugivore interaction dissimilarity (6.41%), followed by the unique contributions of each of these two variables (ecoregion boundaries = 4.22%; spatial distance = 1.90%; Fig. 4). This relatively high contribution of both ecoregion and spatial distance indicates that gradual increases in interaction dissimilarity over space are made significantly steeper when crossing ecoregion boundaries.

## Network structural dissimilarity

Despite significant turnover in species and interaction composition, structural dissimilarity of frugivory networks did not change significantly across large-scale ecological boundaries and human disturbance gradients, being only affected by spatial distance ($F = 20.408$; $P = 0.021$) and differences in sampling intensity ($F = 238.987$; $P = 0.002$) (Supplementary Table 7). These findings held true when evaluating both the binary and quantitative versions of ecoregion and biome distance matrices (Supplementary Tables 7 and 8).

All our main results were robust to different processes of assigning uniqueness to problematic species in local networks, that is, species without a valid epithet that could not be considered as unique species in our dataset (see Supplementary Methods and Supplementary Tables 9–32). Finally, tests of our key hypotheses were not affected by the removal of individual studies (Supplementary Figs. 15 and 16; Supplementary Tables 33 and 34) or small networks (i.e., up to 10 species) from the dataset (see Sensitivity analysis section in the Supplementary Methods).

## Discussion

Our results support the hypothesis that large-scale ecological boundaries drive abrupt differences in species and interaction composition of avian frugivory networks. Specifically, on top of the gradual effect of spatial distance on interaction dissimilarity (whereby networks >2500 km apart had very few interactions in common), transitions across ecoregions and biomes promoted divergence in species interactions. These results show that ecoregions and biomes, classically defined based on environmental conditions and species occurrences[1,3,4], also carry a signature within biotic interactions. Indeed, because the large-scale distribution of both species and interactions is punctuated by ecoregion and biome boundaries (Fig. 2 and Supplementary Fig. 17), our findings suggest that species biogeography is matched by a higher-order biogeography of interactions. In parallel, differences in human disturbance led sites to have significantly different species and interaction composition, which might be partly attributed to the filtering of sensitive species and their interactions from disturbed sites[17,31]. In fact, while networks from natural ecosystems usually contain interactions between native species, which better reflect natural biogeographic patterns[12] and are more susceptible to human disturbances[31], interactions from high-disturbance regions are generally performed by generalist and introduced species[17,31,32]. Nevertheless, despite these differences in composition, we found that the structure of avian frugivory networks was relatively consistent across large-scale environmental gradients. Similar results have been reported at smaller spatial scales[32], indicating that assembly rules may generate common structural patterns in plant-frugivore networks[33] despite the shifts in species and interaction composition that usually accompany environmental changes[15].

Because most of the variation in interaction dissimilarity across biome borders can be explained by ecoregion boundaries, preserving the distinctness of ecoregions[3,4] will likely contribute to maintaining the natural barriers that limit the spread of disturbances across the global network of frugivory. Unfortunately, the unique species assemblages that comprise ecoregions have been increasingly threatened by global changes[3,5]. In fact, the global frugivory network is connected not only through natural processes, such as bird migration[34], but also through human-related processes. Biotic homogenization, in particular, has contributed to blurring biogeographical signatures[12,20] and mitigating the effect of spatial processes on interaction dissimilarity[12]. This notion is reinforced by the fact that all long-distance (>10,000 km) connections (shared interactions) between local networks of frugivory involved at least one region where novel interactions performed by introduced species have largely replaced those performed by declining or already extinct native species, such as Aotearoa New Zealand and Hawai'i[32,35] (see, for example, the shared

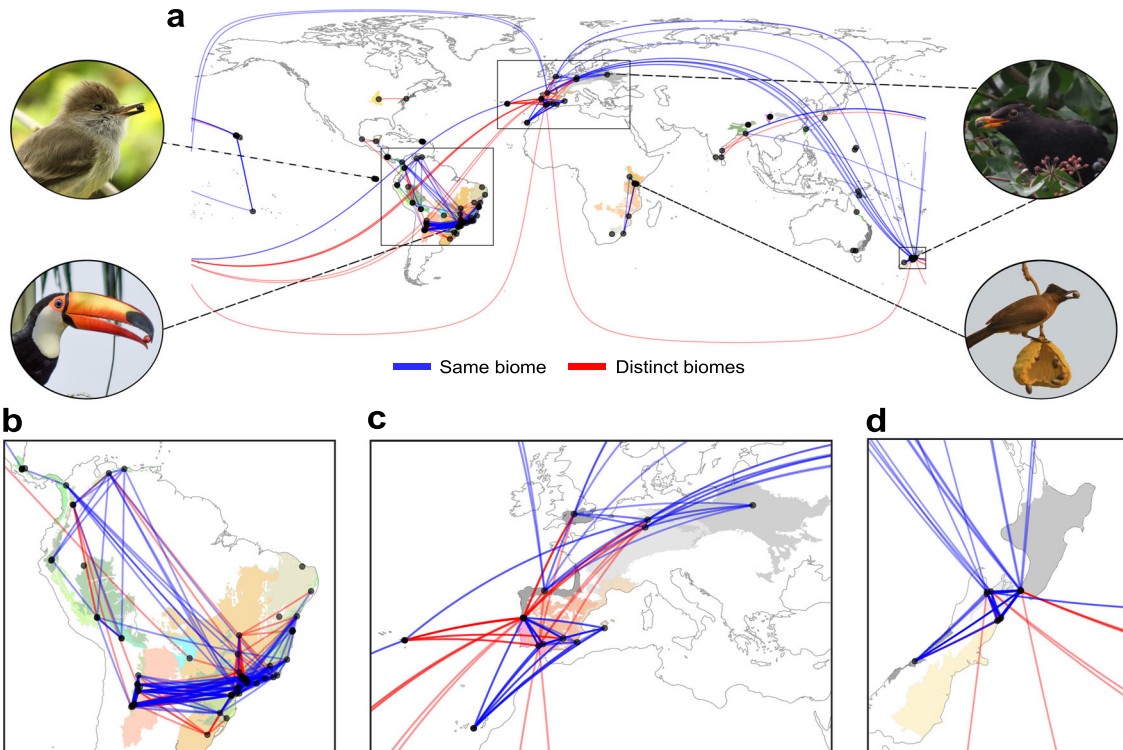

**Fig. 2 | Plant–frugivore interactions shared among local networks, ecoregions, and biomes. a** World map with points representing the 196 avian frugivory networks in our dataset. Colors of shaded areas represent the 67 ecoregions where networks were located, with similar colors indicating ecoregions that belong to the same biome. Lines represent the connections (shared interactions) plotted along the great circle distance between networks, with most of these connections occurring within (blue lines) rather than across (red lines) biomes. Stronger color tones of lines indicate higher similarity of interactions (1-$\beta_{WN}$) between networks. Connections across continents were mostly attributed to introduced species in one of these regions. Lines disappearing at the side edges of the world map are connected to those from the opposite edge. Photos show some of the frugivorous birds present in our dataset. Inset maps depict three regions with many networks and connections (especially within biomes). **b** South America. **c** Europe. **d** Aotearoa New Zealand. Photo credits: R. Heleno (top left and bottom right); R. B. Missano (bottom left); J. M. Costa (top right). Ecoregions and biomes were defined based on the map developed by Dinerstein et al.[3] (available at https://ecoregions.appspot.com/ under a CC-BY 4.0 license). Source data are provided as a Source Data file.

interactions connecting networks from Europe and Aotearoa New Zealand in Fig. 2). Interestingly, these long-distance connections tend to occur more frequently within than across biomes, despite a greater proportion of network comparisons being cross-biome (Supplementary Fig. 18). This indicates that biomes may represent meaningful boundaries not only for species, but also for novel interactions resulting from species introductions around the world[12]. Notably, because species interactions provide the pathways across which direct and indirect effects (such as dynamic impacts of population declines, apparent competition and trophic cascades) may propagate, spatially-separated networks that share interactions may have coupled dynamics and respond similarly to disturbance[9,36]. In fact, findings that ecological networks in adjacent habitats may function as a single dynamic unit[9] raises questions around the scale over which two networks can be considered truly distinct. As a step to answering this question, we provide empirical evidence for the existence of large-scale boundaries between ecological networks. Consequently, our results suggest that disturbances in local frugivory networks are much less likely to impact networks from distant sites or elevations, especially if they are located within distinct ecoregions and biomes.

Although species turnover and interaction dissimilarity responded to similar ecological drivers, species might interact differently across environmental gradients not only because of changes in species composition, but also because of partner switching associated with shifts in species abundance (i.e., the probability of random encounters), foraging behavior and co-evolutionary patterns[15]. Indeed, while interactions necessarily differ when the species involved differ[27], it is possible that shared species interact differently across sites, potentially decoupling the relationship between species turnover and interaction dissimilarity. To evaluate whether interaction rewiring (i.e., the extent to which shared species interact differently[27]) increases across large-scale environmental gradients, we used data limited to pairs of networks sharing plant and bird species (*N* pairs of networks = 1314) (see Rewiring analysis section in "Methods"). We found that interaction rewiring increased significantly across human disturbance, spatial, and elevational gradients (Supplementary Table 35), partially explaining why interactions tend to turn over faster than species at large spatial scales (Supplementary Figs. 9d and 14c). In fact, networks shared considerably more species than interactions (Fig. 2 and Supplementary Fig. 17), reinforcing previous findings that plant and bird species are flexible and tend to switch among their potential partners, even when networks have similar species composition[32]. Surprisingly, we did not find an effect of ecoregion boundaries on interaction rewiring (Supplementary Table 35). This effect only became significant when ecoregion and biome distances were the only predictors in the model (Supplementary Table 36), probably because of their collinearity with our other predictor variables (Supplementary Fig. 19).

As with other large-scale studies of ecological networks[12,37], our data were not evenly spread across the globe, which likely affected the observed patterns. For instance, around 59% of our networks were located within a single biome–the Tropical & Subtropical Moist Broadleaf Forests (Supplementary Fig. 2). Because ecoregions tend to be more distinct in tropical than in temperate zones[38], the greater number of networks from tropical ecosystems (which also possess

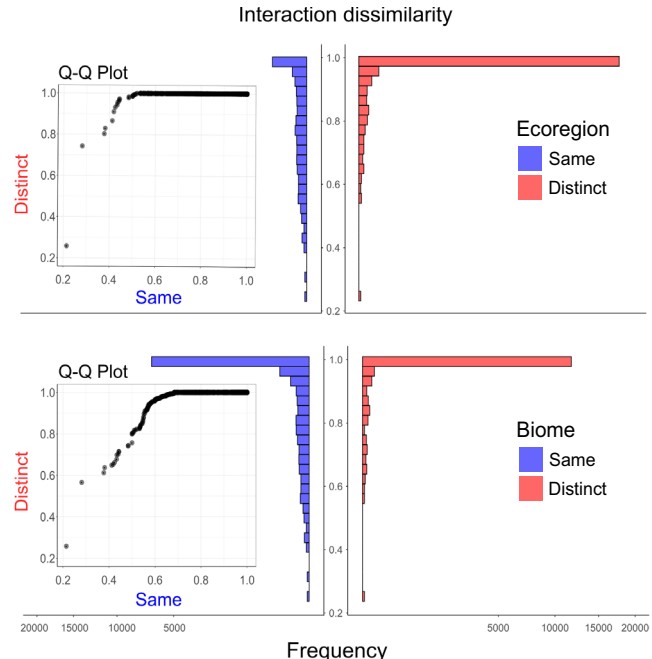

**Fig. 3 | The effects of ecological boundaries on interaction dissimilarity ($\beta_{WN}$).** Histograms and inset quantile-quantile (Q–Q) plots showing differences in the distributions of interaction dissimilarity values between pairs of networks located within ("same") and across ("distinct") ecoregions and biomes. The effects of ecoregion and biome boundaries were significant, even after controlling for the other predictor variables in the model (Table 1). We square root transformed the x-axis scale to allow a better visualization of the distribution of data points (pairs of networks) with interaction dissimilarity values <1. Source data are provided as a Source Data file.

most of the world's ecoregions[3]) may have contributed to the strong observed effect of ecoregion boundaries on interaction dissimilarity. Nevertheless, both species richness and the proportion of frugivorous birds reach their peaks in the Tropics[39], suggesting that the distribution of networks in our dataset partially mirrors the global distribution of avian frugivory. We also highlight that the ecoregions and biomes represented in our dataset cover around 20% and 69%, respectively, of the world's ice-free land surface. As such, network sampling in data deficient regions[37], especially at the ecoregion scale, may contribute greatly to our understanding of macroecological patterns in avian frugivory networks. Importantly, the extent to which our results apply for other frugivorous taxa (such as mammals and reptiles) and interaction types remains to be investigated. Previous findings, however, indicate that less-mobile taxa tend to show a stronger adherence to ecological boundaries[38], a pattern that is likely to be reflected in species interactions. This is corroborated here by the fact that networks located at distinct ecoregions and biomes tended to share more bird than plant species (Supplementary Fig. 17).

This work provides evidence that ecological boundaries and human disturbance gradients delineate the large-scale spatial distribution of species and their interactions. Nevertheless, network structure remained relatively consistent across broad-scale environmental gradients. This suggests that the processes underlying the architecture of frugivory networks, such as ecological specialization[40] and species' functional roles[41], may be reasonably independent of the identity of interacting species[19]. By demonstrating the validity of the ecoregion-based approach[1,3] for species interactions, our results have important implications for maintaining the world's biodiversity of interactions and the myriad ecological functions they provide.

## Methods

### Dataset acquisition

Plant-frugivore network data were obtained through different online sources and publications (Supplementary Table 1). Only networks that met the following criteria were retrieved: (i) the network contains quantitative data (a measure of interaction frequency) from a location, pooling through time if necessary; (ii) the network includes avian frugivores. Importantly, we removed non-avian frugivores from our analyses because only 28 out of 196 raw networks (before data cleaning) sampled non-avian frugivores, and not removing non-avian frugivores would generate spurious apparent turnover between networks that did vs. did not sample those taxa. In addition, the removal of non-avian frugivores did not strongly decrease the number of frugivores in our dataset (Supplementary Fig. 20a) or the total number of links in the global network of frugivory (Supplementary Fig. 20b). Furthermore, non-avian frugivores, as well as their interactions, were not shared across ecoregions and biomes (Supplementary Fig. 21), so their inclusion would only strengthen the results we found (though as noted above, we believe that this would be spurious because they are not as well sampled); (iii) the network (after removal of non-avian frugivores) contains greater than two species in each trophic level. Because this size threshold was somewhat arbitrary, we used a sensitivity analysis to assess the effect of our network size threshold on the reported patterns (see Sensitivity analysis section in the Supplementary Methods and Supplementary Figs. 22–24); and (iv) network sampling was not taxonomically restricted, that is, sampling was not focused on a specific taxonomic group, such as a given plant or bird family. Note, however, that authors often select focal plants or frugivorous birds to be sampled, but this was not considered as a taxonomic restriction if plants and birds were not selected based on their taxonomy (e.g., focal plants were selected based on the availability of fruits at the time of sampling, or focal birds were selected based on previous studies of bird diet in the study site). The first source for network data was the Web of Life database[42], which contains 33 georeferenced plant-frugivore networks from 28 published studies, of which 12 networks met our criteria.

We also accessed the Scopus database on 04 May 2020 using the following keyword combination: ("plant-frugivore*" OR "plant-bird*" OR "frugivorous bird*" OR "avian frugivore*" OR "seed dispers*") AND ("network*" OR "web*") to search for papers that include data on avian frugivory networks. The search returned a total of 532 studies, from which 62 networks that met the above criteria were retrieved. We also contacted authors to obtain plant-frugivore networks that were not publicly available, which provided us a further 110 networks. The remaining networks ($N = 12$) were obtained by checking the database from a recently published study[12]. In total, 196 quantitative avian frugivory networks were used in our analyses.

### Generating the distance matrices to serve as predictor and response variables

**Ecoregion and biome distances.** We used the most up-to-date (2017) map of ecoregions and biomes[3], which divides the globe into 846 terrestrial ecoregions nested within 14 biomes, to generate our ecoregion and biome distance matrices. Of these, 67 ecoregions and 11 biomes are represented in our dataset (Supplementary Figs. 1 and 2). We constructed two alternative versions of both the ecoregion and biome distance matrices. In the first, binary version, if two ecological networks were from localities within the same ecoregion/biome, a dissimilarity of zero was given to this pair of networks, whereas a dissimilarity of one was given to a pair of networks from distinct ecoregions/biomes (this is the same as calculating the Euclidean distance on a presence–absence matrix with networks in rows and ecoregion/biomes in columns).

In the second, quantitative version, we estimated the pairwise environmental dissimilarity between our ecoregions and biomes using

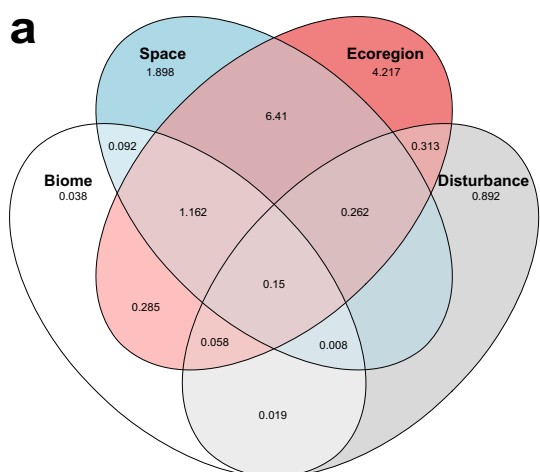

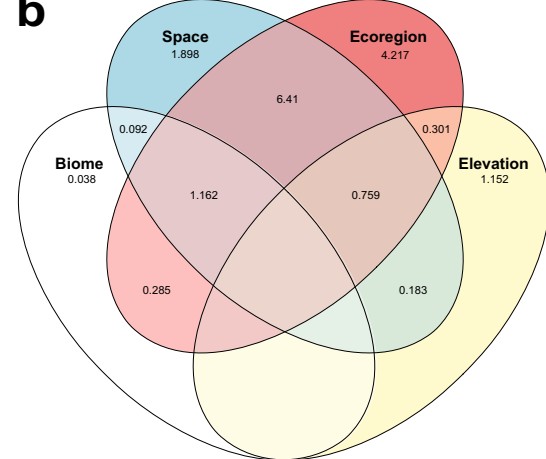

**Fig. 4 | Venn diagrams showing the relative contributions (%) of our main predictor variables to explaining the variation in interaction dissimilarity ($\beta_{WN}$), calculated using deviance partitioning.** Overlapping areas represent deviance that is jointly explained by one or more predictor variables. **a** The relative contributions of ecoregion, biome, spatial and human disturbance (i.e., footprint) distances. In **b**, we replace human disturbance distance with elevational difference;

we show these two separate diagrams for visualization purposes, but Supplementary Fig. 12 shows the effect of all our main predictor variables together. Note that we only plot our predictor variables of interest (i.e., not those used for controlling sampling effects). Terms that reduce explanatory power are not shown. Source data are provided as a Source Data file.

six environmental variables recently demonstrated to be relevant in predicting ecoregion distinctness, namely mean annual temperature, temperature seasonality, mean annual rainfall, rainfall seasonality, slope and human footprint[38]. We obtained climatic and elevation data from WorldClim 2.1[43] at a spatial resolution of 1-km². We transformed the elevation raster into a slope raster using the *terrain* function from the raster package[44] in R[45]. As a measure of human disturbance, we used human footprint—a metric that combines eight variables associated with human disturbances of the environment: the extent of built environments, crop land, pasture land, human population density, night-time lights, railways, roads and navigable waterways[26]. The human footprint raster was downloaded at a 1-km² resolution[26]. Because human footprint data were not available for one of our ecoregions (Galápagos Islands xeric scrub), we estimated human footprint for this ecoregion by converting visually interpreted scores into the human footprint index. We did this by analyzing satellite images of the region and following a visual score criterion[26]. Given the previously demonstrated strong agreement between visual score and human footprint values[26], we fitted a linear model using the visual score and human footprint data from 676 validation plots located within the Deserts and xeric shrublands biome - the biome in which the Galápagos Islands xeric scrub ecoregion is located - and estimated the human footprint values for our own visual scores using the *predict* function in R[45].

We used 1-km² resolution rasters and the *extract* function from the raster package[44] to calculate the mean value of each of our six environmental variables for each ecoregion in our dataset. Because biomes are considerably larger than ecoregions (which makes obtaining environmental data for biomes more computationally expensive) we used a coarser spatial resolution of 5-km² for calculating the mean values of environmental variables for each biome. Since a 5-km² resolution raster was not available for human footprint, we transformed the 1-km² resolution raster into a 5-km² raster using the *resample* function from the same package.

To combine these six environmental variables into quantitative matrices of ecoregion and biome environmental dissimilarity, we ran a Principal Component Analysis (PCA) on our scaled multivariate data matrix (where rows are ecoregions or biomes and columns are environmental variables). From this PCA, we selected the scores of the four and three principal components, which

represented 89.6% and 88.7% of the variance for ecoregions and biomes, respectively, and converted it into a distance matrix by calculating the Euclidean distance between pairs of ecoregions/biomes using the *vegdist* function from the vegan package[46]. Finally, we transformed the ecoregion or biome distance matrix into a $N \times N$ matrix where $N$ is the number of local networks. In this matrix, cell values represent the pairwise environmental dissimilarity between the ecoregions/biomes where the networks are located. The main advantage of using this quantitative approach is that, instead of simply evaluating whether avian frugivory networks located in distinct ecoregions or biomes are different from each other in terms of network composition and structure (as in our binary approach), we were also able to determine whether the extent of network dissimilarity depended on how environmentally different the ecoregions or biomes are from one another.

**Local-scale human disturbance distance.** To generate our local human disturbance distance matrix, we extracted human footprint data at a 1-km² spatial resolution[26] and calculated the mean human footprint values within a 5-km buffer zone around each network site. For the networks located within the Galápagos Islands xeric scrub ecoregion ($N = 4$), we estimated the human footprint index using the same method described in the previous section for ecoregion- or biome-scale human footprint. We then calculated the pairwise Euclidean distance between human footprint values from our network sites. Thus, low cell values in the local human disturbance distance matrix indicate pairs of network sites with a similar level of human disturbance, while high values represent pairs of network sites with very different levels of human disturbance.

**Spatial distance.** The spatial distance matrix was generated using the Haversine (i.e., great circle) distance between all pairwise combinations of network coordinates. In this matrix, cell values represent the geographical distance between network sites.

**Elevational difference.** We calculated the Euclidean distance between pairwise elevation values (estimated as meters above sea level) of network sites to generate our elevational difference matrix. Elevation values were obtained from the original sources when available or using Google Earth[47]. In the elevational difference matrix, low cell values

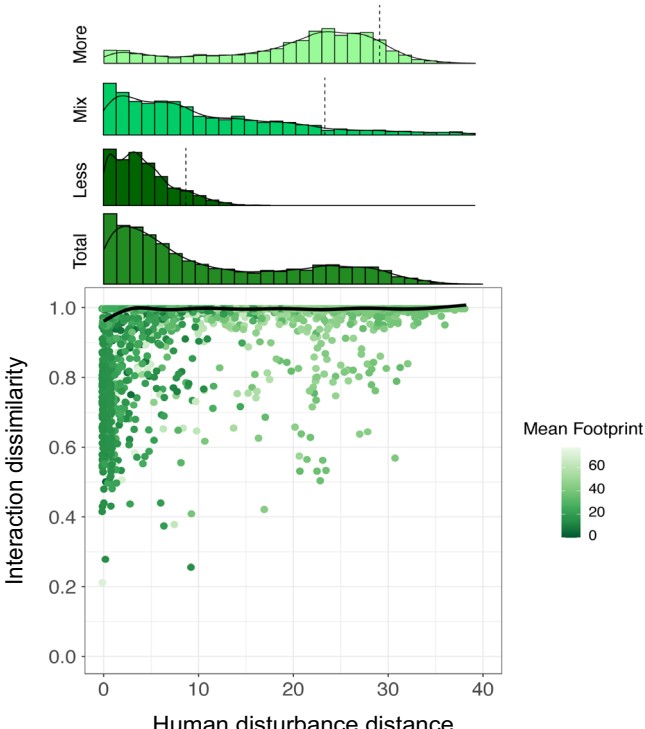

**Fig. 5 | The effect of human disturbance gradients on interaction dissimilarity ($\beta_{WN}$).** The relationship between human disturbance distance and interaction dissimilarity, with a fitted line obtained from a Generalized Additive Model (GAM) with human disturbance distance as the only predictor variable (Supplementary Fig. 13 shows the partial effects plot for the model including all predictors). Human disturbance distance was calculated as the absolute difference in human footprint values between a pair of network sites. Each data point (pair of networks) is colored according to the mean of the human footprint values from the two networks. The histogram above the plot shows the distribution of data points across the human disturbance gradient. To explore whether disturbance distance and the mean intensity of disturbance are related, we further divided our data into three equal sized groups (top three histograms) based on their mean (of the site pair) footprint values: "Less" disturbed (low mean footprint), 'Mix' (medium mean footprint) and 'More' disturbed (high mean footprint). Dashed lines mark the 90th percentile position in each histogram. Note that data points from less disturbed site pairs are skewed towards low values of human disturbance distance, whereas pairs of more disturbed sites also had a larger average distance. Source data are provided as a Source Data file.

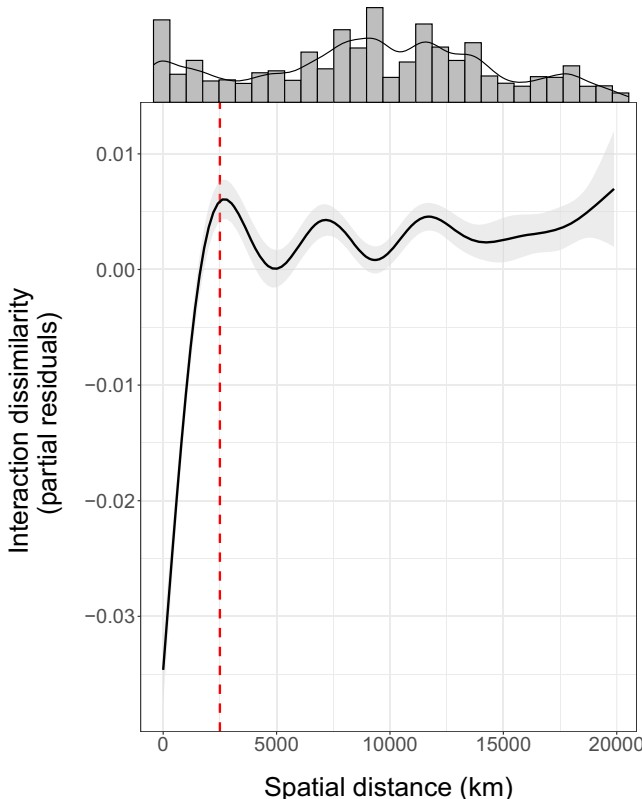

**Fig. 6 | Partial effects plot of the relationship between spatial distance and interaction dissimilarity ($\beta_{WN}$).** Here, we show the fit (solid line) of a Generalized Additive Model (GAM) with interaction dissimilarity as the response variable and all our predictor variables included. Thus, this plot shows the effect of spatial distance on interaction dissimilarity, while controlling for the effect of the other predictor variables in the model. Partial residuals remain on the same scale as the original data, but the sign of values indicates how they differ from what would be expected (i.e., from the fitted values) based on the other predictor variables in the model. The gray area represents two standard errors above and below the estimate of the smooth being plotted. The histogram above the plot shows the distribution of data points across the spatial gradient. Note the sharp increase in interaction dissimilarity until a threshold distance of around 2500 km (dotted red line), beyond which few networks shared interactions (a similar pattern can be seen in Supplementary Fig. 14c). Source data are provided as a Source Data file.

represent pairs of network sites within similar elevations, whereas high values represent pairs of network sites within very different elevations.

**Network sampling dissimilarity.** We used the metadata retrieved from each of our 196 local networks to generate our network sampling dissimilarity matrices, which aim to control statistically for differences in network sampling. There are many ways in which sampling effort could be quantified, so we began by calculating a variety of metrics, then narrowed our options by assessing which of these was most related to network metrics. We divided the sampling metrics into two categories: time span-related metrics (i.e., sampling hours and months) and empirical metrics of sampling completeness (i.e., sampling completeness and sampling intensity), which aim to account for how complete network sampling was in terms of species interactions (Supplementary Table 2).

We selected the quantitative sampling metrics to be included in our models based on (i) the fit of generalized linear models evaluating the relationship between number of sampling hours and sampling months of the study and network-level metrics (i.e., bird richness, plant richness and number of links), and (ii) how well time span-related

metrics, sampling completeness and sampling intensity predicted the proportion of known interactions that were sampled in each local network (hereafter, ratio of interactions) for a subset of the data. This latter metric, defined as the ratio between the number of interactions in the local network and the number of known possible interactions in the region involving the species in the local network, captures raw sampling completeness. Therefore, ratio of interactions estimates, for a given set of species, the proportion of all their interactions known for a region that are found to occur among those same species in the local network. To calculate this metric, we needed high-resolution information on the possible interactions, so we used a subset of 14 networks sampled in Aotearoa New Zealand, since there is an extensive compilation of frugivory events recorded for this country[48]. After this process, we selected number of sampling hours, number of sampling months and sampling intensity for inclusion in our statistical models (Supplementary Figs. 7 and 8; Supplementary Table 2). We generated the corresponding distance matrices by calculating the Euclidean distance between metric values. Similarly, we generated a Euclidean distance matrix for differences in sampling year between pairs of networks, which aims to account for long-term changes in the environment, species composition and network sampling methods. We obtained the sampling year of our local networks from the original

sources and calculated the mean sampling year value for those networks sampled across multiple years.

Because sampling methods, such as sampling design, focus (i.e., focal taxa, which determines whether a zoocentric or phytocentric method was used), interaction frequency type (i.e., how interaction frequency was measured) and coverage (total or partial) might also affect the observed plant-frugivore interactions[49], we combined these variables into a single distance matrix to estimate the overall differences in sampling methods between networks. Because most of these variables were categorical with multiple levels (Supplementary Table 3), we generated our method's dissimilarity matrix by using a generalization of Gower's distance method[50], which allows the treatment of different types of variables when calculating distances. For this, we used the *dist.ktab* function from the ade4 package[51]. We ran a Principal Coordinates Analysis (PCoA) on this distance matrix, selected the first four axes, which explained 81.2% of the variation in method's dissimilarity, and calculated the Euclidean distance between pairs of networks using the *vegdist* function from the vegan package[46] in R[45].

**Network dissimilarity.** We generated three network dissimilarity matrices to be our response variables in the statistical models. In the first, cell values represent the pairwise dissimilarity in species composition between networks (beta diversity of species; $\beta_S$)[27]. Second, we measured interaction dissimilarity (beta diversity of interactions; $\beta_{WN}$), which represents the pairwise dissimilarity in the identity of interactions between networks[27]. Importantly, we did not include interaction rewiring ($\beta_{OS}$) in our main analysis because this metric can only be calculated for networks that share interaction partners (i.e., it estimates whether shared species interact differently)[27], which limited the number and the spatial distribution of networks available for analysis (but see the Rewiring analysis section for an analysis on the subset of our dataset for which this was possible). Metrics were calculated using the *network_betadiversity* function from the betalink package[52] in R[45].

Finally, we calculated a third dissimilarity matrix to capture overall differences in network structure. We recognize that there are many potential metrics of network structure, and that many of these are strongly correlated with one another[53–56]. We therefore chose a range of metrics that captured the number of links, their relative weightings (including across trophic levels), and their arrangement among species, then combined these into a single distance matrix. Specifically, we quantified network structural dissimilarity using the following metrics: weighted connectance, weighted nestedness, interaction evenness, PDI and modularity.

Weighted connectance represents the number of links relative to the number of possible links, weighted by the frequency of each interaction[55], and is therefore a measure of network-level specialization (higher values of weighted connectance indicate lower specialization). Importantly, it has been suggested that connectance affects persistence in mutualistic systems[54]. We measured nestedness (i.e., the pattern in which specialist species interact with proper subsets of the species that generalist species interact with) using the weighted version of nestedness based on overlap and decreasing fill (wNODF)[57]. Notably, nested structures have been commonly reported in plant-frugivore networks[33]. Interaction evenness is Shannon's evenness index applied for species interactions and represents how evenly distributed the interactions are in the network[21,58]. This metric has been previously demonstrated to decline with habitat modification as a consequence of some interactions being favored over others in high-disturbance environments[21]. PDI (Paired Difference Index) is a measure of species-level specialization on resources and a reliable indicator not only of specialization, but also of absolute generalism[59]. Thus, this metric contributes to understanding of the ecological processes that drive the prevalence of specialists or generalists in ecological networks[59]. In order to obtain a network-level PDI, we calculated the weighted mean PDI for each local network. Finally, we calculated

modularity (i.e., the level of compartmentalization within networks) using the DIRTPLAwb+ algorithm[60]. Modularity estimates the extent to which species within modules interact more with each other than with species from other modules[61], and it has been demonstrated to affect the persistence and resilience of mutualistic networks[54]. All the selected network metrics are based on weighted (quantitative) interaction data, as these have been suggested to be less biased by sampling incompleteness[62] and to better reflect environmental changes[21]. All network metrics were calculated using the bipartite package[63] in R[45].

We ran a Principal Component Analysis (PCA) on our scaled multivariate data matrix ($N \times M$ where $N$ is the number of local networks in our dataset and $M$ is the number of network metrics), selected the scores of the three principal components, which represented 89.9% of the variance in network metrics, and converted it into a network structural dissimilarity matrix by calculating the Euclidean distance between networks. In this distance matrix, cell values represent differences in the overall architecture of networks (over all the network metrics calculated), and therefore provide a complementary approach for evaluating how species interaction patterns vary across large-scale environmental gradients.

### Statistical analysis

We employed a two-tailed statistical test that combines Generalized Additive Models (GAM)[29] and Multiple Regression on distance Matrices (MRM)[30] to evaluate the effect of each of our predictor distance matrices on our response matrix. With this approach, we were able to fit GAMs where the predictor and responsible variables are distance matrices, while accounting for the non-independence of distances from each local network by permuting the response matrix[30]. The main advantage of using GAMs is their flexibility in modeling non-linear relationships through smooth functions, which are represented by a sum of simpler, fixed basis functions that determine their complexity[29]. Using GAM-based MRM models allowed us to obtain $F$ values for each of the smooth terms (i.e., smooth functions of the predictor variables in our model), and test statistical significance at the level of individual variables. The binary versions of ecoregion and biome distance matrices (with two levels, "same" or "distinct") were treated as categorical variables in the models, and $t$ values were used for determining statistical significance. We fitted GAMs with thin plate regression splines[64] using the *gam* function from the mgcv package[29] in R[45]. Smoothing parameters were estimated using restricted maximum likelihood (REML)[29]. Our GAM-based MRM models were calculated using a modified version of the *MRM* function from the ecodist package[65], which allowed us to combine GAMs with the permutation approach from the original *MRM* function (see Code availability). All the models were performed with 1000 permutations (i.e., shuffling) of the response matrix.

We explored the unique and shared contributions of our predictor variables to network dissimilarity using deviance partitioning analyses. These were performed by fitting reduced models (i.e., GAMs where one or more predictor variables of interest were removed) using the same smoothing parameters as in the full model and comparing the explained deviance. We fixed smoothing parameters for comparisons in this way because these parameters tend to vary substantially (to compensate) if one of two correlated predictors is dropped from a GAM.

### Assessing the influence of individual studies on the reported patterns

Because our dataset comprises 196 local frugivory networks obtained from 93 different studies, and some of these studies contained multiple networks, we needed to evaluate whether our results were strongly biased by individual studies. To do this, we followed the approach from a previous study[66] and tested whether $F$ values of smooth terms and $t$ values of categorical variables (binary version of

ecoregion and biome distances) changed significantly when jack-knifing across studies. We did this by dropping one study from the dataset and re-fitting the models, and then repeating this same process for all the studies in our dataset.

We found a number of consistent patterns within different subsets of the data (Supplementary Figs. 15 and 16); however, some of the patterns we observed appear to be driven by individual studies with multiple networks, and hence are less representative. For instance, the study with the greatest number of networks in our dataset (study ID = 76), which contains 35 plant-frugivore networks sampled across an elevation gradient in Mt. Kilimanjaro, Tanzania[67], had an overall high influence on the results when compared with the other studies. By re-running our GAM-based MRM models after removing this study from our dataset, we found that the effect of biome boundaries on interaction dissimilarity is no longer significant, whereas the effects of ecoregion boundaries, human disturbance distance, spatial distance and elevational differences remained consistent with those from the full dataset (Supplementary Table 33). Nevertheless, all the results were qualitatively similar to those obtained for the entire dataset when using network structural dissimilarity as the response variable (Supplementary Table 34).

### Rewiring analysis

Interaction rewiring ($\beta_{OS}$) estimates the extent to which shared species interact differently[27]. Because this metric can only be calculated for networks that share species from both trophic levels, we selected a subset of network pairs that shared plants and frugivorous birds ($N = 1314$) to test whether interaction rewiring increases across large-scale environmental gradients. Importantly, since not all possible combinations of network pairs contained values of interaction rewiring (i.e., not all pairs of networks shared species), a pairwise distance matrix could not be generated for this metric. Thus, we were not able to use the same statistical approach used in our main analysis, which is based on distance matrices (see Statistical analysis section). Instead, we performed a Generalized Additive Mixed-effects Model (GAMM) using ecoregion, biome, human disturbance, spatial, elevational, and sampling-related distance metrics as fixed effects and network IDs as random effects (to account for the non-independence of distances) (Supplementary Table 35). We also performed a reduced model with only ecoregion and biome distance metrics as predictor variables (Supplementary Table 36). The binary version of ecoregion and biome distance metrics (with two levels, "same" or "distinct") were used as categorical variables in both models. Interaction rewiring ($\beta_{OS}$) was calculated using the *network_betadiversity* function from the betalink package[52] in R[45]. Although it has been recently argued that this metric may overestimate the importance of rewiring for network dissimilarity[68], our main focus was not the partitioning of network dissimilarity into species turnover and rewiring components, but rather simply detecting whether the sub-web of shared species interacted differently. In this case, $\beta_{OS}$ (as developed by ref. 27) is an adequate and useful metric[68]. We fitted our models using the *gamm4* function from the gamm4 package[69] in R[45]. Smoothing parameters were estimated using restricted maximum likelihood (REML)[29].

### Reporting summary

Further information on research design is available in the Nature Research Reporting Summary linked to this article.

## Data availability

The data necessary to reproduce the analyses of this manuscript have been deposited in the Dryad database: https://doi.org/10.5061/dryad.mcvdnck4d (ref. 70). Metadata of the plant-frugivore networks, and predictor and response variables used in our analyses are provided with this paper as Supplementary Data. The Ecoregions 2017 © Resolve map developed by ref. 3 is available at https://ecoregions.appspot.

com/ under a CC-BY 4.0 license. Human footprint data are publicly available at https://doi.org/10.5061/dryad.052q5 (ref. 71). The World-Clim 2.1 database[43] is publicly available at https://www.worldclim.org/. The following taxonomic databases were used for standardizing the taxonomy of plant and bird species in our dataset: Global Names Resolver (GNR) (available at https://resolver.globalnames.org/), National Center for Biotechnology Information (NCBI) (available at https://ncbi.nlm.nih.gov/), BirdLife International (available at http://datazone.birdlife.org/species/taxonomy), Avibase (available at https://avibase.bsc-eoc.org/), Integrated Taxonomic Information System (ITIS) (available at https://itis.gov/), International Plant Names Index (IPNI) (available at https://www.ipni.org/), Tropicos (available at https://www.tropicos.org/), and the iPlant Taxonomic Name Resolution Service[72] (available at https://tnrs.biendata.org/). Source data are provided with this paper.

## Code availability

R scripts for reproducing the analyses of this manuscript are available at https://doi.org/10.5061/dryad.mcvdnck4d (ref. 70).

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

## Acknowledgements

We thank all the researchers in Tylianakis and Stouffer lab groups for their insightful comments on this manuscript. The authors acknowledge the following funding: University of Canterbury Doctoral Scholarship (L.P.M.); The Marsden Fund grant UOC1705 (J.M.T., L.P.M.); The São Paulo Research Foundation - FAPESP 2014/01986-0 (M.G., C.E.), 2015/15172-7 and 2016/18355-8 (C.E.), 2004/00810-3 and 2008/10154-7 (C.I.D., M.G., M.A.P.); Earthwatch Institute and Conservation International for financial support (C.I.D., M.G., M.A.P.); Carlos Chagas Filho Foundation for Supporting Research in the Rio de Janeiro State – FAPERJ grant E-26/200.610/2022 (C.E.); Brazilian Research Council grants 540481/01-7 and 304742/2019-8 (M.A.P.) and 300970/2015-3 (M.G.); Rufford Small Grants for Nature Conservation No. 22426–1 (J.C.M., I.M.), No. 9163-1 (G.B.J.) and No. 11042-1 (MCM); Universidade Estadual de Santa Cruz (Propp-UESC; No. 00220.1100.1644/10-2018) (J.C.M., I.M.); Fundação de Amparo à Pesquisa do Estado da Bahia - FAPESB (No. 0525/2016) (J.C.M., I.M.); European Research Council under the European Union's Horizon 2020 research and innovation program (grant 787638) and The Swiss National Science Foundation (grant 173342), both awarded to C. Graham (D.M.D.); ARC SRIEAS grant SR200100005 Securing Antarctica's Environmental Future (D.M.D.); German Science Foundation —Deutsche Forschungsgemeinschaft PAK 825/1 and FOR 2730 (K.B.G., E.L.N., M.Q., V.S., M.S.), FOR 1246 (K.B.G., M.S., M.G.R.V.) and HE2041/20-1 (F.S., M.S.); Portuguese Foundation for Science and Technology - FCT/MCTES contract CEECIND/00135/2017 and grant UID/BIA/04004/2020 (S.T.) and contract CEECIND/02064/2017 (L.P.S.); National Scientific and Technical Research Council, PIP 592 (P.G.B.); Instituto Venezolano de Investigaciones Científicas - Project 898 (V.S.D.).

## Author contributions

Conceptualization: L.P.M. and J.M.T.; Methodology: L.P.M., J.M.T., and D.B.S.; Collection of data: P.G.B., K.B.G., G.B.J., M.C., J.M.C., D.M.D., C.I.D., C.E., M.G., R.H., P.J., I.M., J.C.M., M.C.M., E.L.N., M.A.P., M.Q., R.A.R., F.S., V.S., V.S.D., M.S., L.P.S., F.R.S., S.T., A.T., M.G.R.V.; Writing of original draft: L.P.M. and J.M.T. All authors contributed to the final version of the manuscript.

## Competing interests

The authors declare no competing interests.

## Additional information

**Lucas P. Martins** [1] ✉, **Daniel B. Stouffer** [1], **Pedro G. Blendinger** [2,3], **Katrin Böhning-Gaese** [4,5], **Galo Buitrón-Jurado** [6,7], **Marta Correia** [8], **José Miguel Costa** [8], **D. Matthias Dehling** [9,10], **Camila I. Donatti** [11,12], **Carine Emer** [13,14], **Mauro Galetti** [14], **Ruben Heleno** [8], **Pedro Jordano** [15,16], **Ícaro Menezes** [17], **José Carlos Morante-Filho** [17], **Marcia C. Muñoz** [18], **Eike Lena Neuschulz** [4], **Marco Aurélio Pizo** [14], **Marta Quitián** [19,20], **Roman A. Ruggera** [21], **Francisco Saavedra** [22], **Vinicio Santillán** [23], **Virginia Sanz D'Angelo** [6], **Matthias Schleuning** [4], **Luís Pascoal da Silva** [24,25], **Fernanda Ribeiro da Silva** [26], **Sérgio Timóteo** [8], **Anna Traveset** [20], **Maximilian G. R. Vollstädt** [27] & **Jason M. Tylianakis** [1] ✉

[1] Centre for Integrative Ecology, School of Biological Sciences, University of Canterbury, Private bag 4800, Christchurch 8140, Aotearoa New Zealand. [2] Instituto de Ecología Regional, Universidad Nacional de Tucumán and CONICET; CC 34, 4107 Tucumán, Argentina. [3] Facultad de Ciencias Naturales e Instituto Miguel Lillo, Universidad Nacional de Tucumán, Miguel Lillo 2005, 4000 Tucumán, Argentina. [4] Senckenberg Biodiversity and Climate Research Centre (SBiK-F), Senckenberganlage 25, 60325 Frankfurt am Main, Germany. [5] Institute for Ecology, Evolution and Diversity, Goethe University Frankfurt, Max-

von-Laue-Straße 13, Frankfurt am Main 60439, Germany. [6]Laboratorio de Biología de Organismos, Centro de Ecología, Instituto Venezolano de Investigaciones Científicas (IVIC), Carretera Panamericana, km 11, Altos de Pipe, Edo, Miranda, Venezuela. [7]Universidad Estatal Amazónica-Sede Zamora Chinchipe; Calle Luis Imaicela entre Azuay y Rene Ulloa, El Pangui, Zamora Chinchipe, Ecuador. [8]Centre for Functional Ecology, Associate Laboratory TERRA, Department of Life Sciences, University of Coimbra, Calçada Martim de Freitas, 3000-456 Coimbra, Portugal. [9]Swiss Federal Research Institute WSL, Zürcherstrasse 111, 8903 Birmensdorf, Switzerland. [10]Securing Antarctica's Environmental Future, School of Biological Sciences, Monash University, Melbourne, Victoria 3800, Australia. [11]Conservation International, 2011 Crystal Dr. Suite 600, Arlington, VA 22202, USA. [12]Department of Biological Sciences, Northern Arizona University, 617S. Beaver St., Flagstaff, AZ 86011-5640, USA. [13]Rio de Janeiro Botanical Garden Research Institute, Rua Pacheco Leão 915, Jardim Botânico, Rio de Janeiro, RJ CEP 22460-030, Brazil. [14]Department of Biodiversity, São Paulo State University – UNESP, Rio Claro, SP, Brazil. [15]Estación Biológica de Doñana, CSIC, av. Americo Vespucio 26, 41092 Sevilla, Spain. [16]Departamento de Biología Vegetal y Ecología, Universidad de Sevilla, Sevilla, Spain. [17]Applied Conservation Ecology Lab, Santa Cruz State University, Rodovia Ilhéus- Itabuna, km 16, Salobrinho, Ilhéus, Bahia 45662-000, Brazil. [18]Programa de Biología, Universidad de La Salle, Carrera 2 # 10-70 Bogotá, Colombia. [19]Systematic Zoology Laboratory, Tokyo Metropolitan University, 1-1 Minami-Osawa, Hachioji-shi, Tokyo 192-0397, Japan. [20]Instituto Mediterráneo de Estudios Avanzados (CSIC-UIB), Miquel Marqués 21, Mallorca, Balearic Islands, 07190 Esporles, Spain. [21]Instituto de Ecorregiones Andinas (Consejo Nacional de Investigaciones Científicas y Técnicas - Universidad Nacional de Jujuy), Canónigo Gorriti 237, Y4600 San Salvador de Jujuy, Jujuy, Argentina. [22]Instituto de Ecología, Facultad de Ciencias Puras y Naturales, Universidad Mayor de San Andrés, La Paz, Bolivia. [23]Centro de Investigación, Innovación y Transferencia de Tecnología (CIITT), Unidad Académica de Posgrado, Universidad Católica de Cuenca, Av. de las Américas, Cuenca, Ecuador. [24]CIBIO, Centro de Investigação em Biodiversidade e Recursos Genéticos, InBIO Laboratório Associado, Campus de Vairão, Universidade do Porto, 4485-661 Vairão, Portugal. [25]BIOPOLIS Program in Genomics, Biodiversity and Land Planning, CIBIO, Campus de Vairão, 4485-661 Vairão, Portugal. [26]Laboratory of Human Ecology and Ethnobotany, Department of Ecology and Zoology, Federal University of Santa Catarina, UFSC, Campus Trindade, s/n, Florianópolis, SC 88010-970, Brazil. [27]Section for Molecular Ecology and Evolution, Globe Institute, University of Copenhagen, Oester Voldgade 5-7, 1350 Copenhagen K, Denmark. ✉e-mail: martinslucas.p@gmail.com; jason.tylianakis@canterbury.ac.nz

