## [Peer review file · Nature Communications]

REVIEWER COMMENTS

Reviewer #1 (Remarks to the Author):

In the present manuscript the authors study dissimilarity in observed avian frugivory networks and, in particular, the dependence on a variety of factors including biomes, ecotypes, elevation etc.

The authors raise interesting questions and probably invested much effort in assembling and standardizing the database. However from a data science perspective I find this manuscript rather weak. This type of data and is notoriously noisy and hence analysis must control for this. However, the paper just combines several fairly standard procedures and packages. Essential bits such as the distance metrics are ad-hoc defined and don't take advantage of recent insights in network metrics. Analysis of complex relationships by a linear method is questionable and prone to lead to artifacts. The link between numerical results, hypothesis and conclusions is quite weak. The authors consider hypothesis based on structurally different factors (e.g. 1d continuous elevation vs. 2d discrete ecotype) and then evaluate with an approach that is agnostic of these structural differences, which again can easily yield misleading results.

The results of the analysis seem plausible but this alone is insufficient to support the validity of the conclusions.

In summary this the authors investigate an interesting question using an interesting dataset, but their methodology is not too simplistic to convincingly support the conclusions.

Reviewer #2 (Remarks to the Author):

Review of "Global and regional ecological boundaries drive abrupt changes in avian frugivory interactions" by Martins et al.

The authors investigate how the composition and structure of avian frugivory networks varies within and between ecoregions and biomes, and also based on the level of human disturbance. They find that networks in different ecoregions and biomes contain dissimilar sets of species and interactions, with dissimilarity increasing with the level of human disturbance; however, network structure does not change significantly across ecoregions and biomes.

The authors address an interesting question using a great data set and the methods appear sound. However, the presentation needs to be improved: it is hard to follow what's going on regarding the actual analysis unless someone first reads the Methods section. Throughout the Introduction, Results and Discussion, there is too much unexplained jargon and insufficient explanation of methods such that I could not really understand what the authors did or the motivation for the study, nor interpret results. I expand on this below as three major comments: (i) Make the Introduction understandable to a general scientist and provide more description of methods in the main narrative, (ii) Include actual results in the main text, and (iii) Address some technical concerns. The authors have performed a lot of analyses, but it is by-and-large left up to the reader to piece together 23 figures (5 in main text plus 18 in SI) and 38 Tables (1 in main text plus 37 in SI) to understand what was done and what the results are. The work has the potential to be important and impactful, but this requires that the authors more clearly describe their main findings (in more straightforward language) and, more importantly, explain the supporting evidence.

(i) Make the Introduction understandable to a general scientist and provide more description of methods in the main narrative

The Introduction, and especially the presentation of hypotheses, should be presented in a way that is understandable to a general scientist, or at least a typical ecologist. Right now, the audience appears to be limited to ecological network researchers.

The main hypothesis (L107), "that the large-scale distribution of species interactions is punctuated by ecoregion and biome boundaries," is too abstract. What exactly is being tested and why? Even the more specific hypothesis (L127) contains jargon---"interaction dissimilarity," "changes in the whole structure of networks"---that is not defined (nor are the terms necessarily intuitive). What exactly is expected to change: the number and/or identity of interactions, species, value of network metrics?

The authors assume the reader knows what an avian frugivory network is (L124), whereas it would be safer to explain what the nodes and links represent. Similarly, they should explain what weights represent in their quantitative networks (L135).

There also needs to be more description of methods in the main narrative, so that the reader doesn't have to constantly jump back-and-forth to the Methods section (without adequate direction, it must be noted) or SI. Indeed, the first part of only two descriptions of methods in the main narrative invites the reader to look at 6 supplementary figures and 2 supplementary tables to understand how they accounted for sampling effects (L137): "To ensure that our results would not be driven by taxonomic uncertainty and sampling effects, we standardized the taxonomy of plant and bird species in our

networks following a series of steps (Supplementary Figs. 3-6) and controlled statistically for network sampling metrics in our analyses (Supplementary Figs. 7-8; Supplementary Tables 2-3)." Even a few sentences describing what was actually done would be helpful.

Then the second part describing methods is, I feel, insufficient for the reader to form even the vaguest intuition of what analysis was performed (L143): "We generated several distance matrices ($N \times N$, where N is the number of local networks in our dataset) to be our variables in the statistical models. Specifically, we used ecoregion, biome, local human disturbance (measured using the human footprint index), spatial, elevation and sampling-related (i.e., hours, months, years, intensity and methods) distance matrices as predictor variables, and facets of network dissimilarity (i.e., species turnover, interaction dissimilarity, and network structural dissimilarity) as response variables."

Moreover, I don't think the authors satisfactorily explain why studying "network dissimilarity" (their response variable) is interesting in the first place.

(ii) Include actual results in the main text

There should be more numerical results in the main narrative, along with more informative summaries of observed trends. For example, the opening of the Results section simply tells the reader to go look at 3 figures and 2 tables in the SI to get an idea of the actual results (L152): "We found that the turnover of plant and frugivorous-bird species composition was strongly affected by ecoregion and biome boundaries (Supplementary Tables 4 and 5; Supplementary Figs. 9a-b). Similarly, there was an overall trend of networks located at different positions along the human disturbance gradient having different species composition (Supplementary Fig. 9c)." A large part of the issue is that the reader encounters the results without really understanding what analysis was done.

Even 4 of the 5 figures in the main text are covered in only 3 sentences (L165): "This provides strong support to the hypothesis that large-scale ecological boundaries mark spatially abrupt changes in plant-frugivore interactions (Figs. 1 and 2; Supplementary Fig. 11). Importantly, a great proportion of the deviance explained by biomes was shared with ecoregions (Fig. 3; Supplementary Fig. 12), which suggests that changes in interaction dissimilarity across biome boundaries mostly reflect the variation occurring at a finer (ecoregion) scale. In fact, crossing an ecoregion boundary induced an average 7% increase in interaction dissimilarity, while crossing a biome boundary induced only an additional 0.2% change. As with species, networks located at opposite ends of the human disturbance continuum usually exhibited very different interactions (Fig. 4; Supplementary Fig. 13)." Again, at this point in the text, it wasn't clear how to interpret these results, given the so far limited explanation of methods.

Another example is the additional description of Figure 3 (L189): "The shared effect of ecoregion boundaries and spatial distance explained the greatest proportion of the variability in plant-frugivore interaction dissimilarity, followed by the unique contributions of these two variables (Fig. 3)." It is not clear what "shared effect" means, nor "proportion of the variability in plant-frugivore interaction dissimilarity," nor "unique contributions to these two variables"---how is a boundary expressed as a variable?

There are some very interesting observations in the Discussion section (L215 to L221), but supporting evidence needs to be presented, explained, and linked in the main narrative. I would also like to see more discussion on (L248), "these results suggest that disturbances in local networks of frugivory are much less likely to impact networks from distant sites or elevations, especially if the networks are located within distinct ecoregions and biomes;" specifically, regarding the mechanisms that can lead to long-distance effects, i.e., how does one go from: (i) there are two networks separated by a large distance, (ii) one network experiences a disturbance, and (iii) this affects the second network? And (L285), "suggesting that the ecological processes underlying the architecture of frugivory networks may be independent of species and interaction composition"---which processes?

(iii) Address some technical concerns

The authors remove non-avian frugivores from analyses (L297)---how much does this affect results?

The minimum size of network analyzed can be very small (potentially containing only 4 species, L299). It would be informative to see some data/network summary statistics in the main text. I worry that much of the analysis, especially of network properties, is not suitable for such small networks; I would recommend setting a minimum network size to something more like 8 to 12 species---how many networks would that rule out from analysis?

Minor comments

-- L204. Explain, "all our main results were robust to different processes of assigning uniqueness to problematic species (i.e., species without a valid epithet) (Supplementary Tables 9-33)."

-- L241. Explain, "at least one region where novel interactions performed by introduced species have largely replaced those performed by native species..."

-- L320. Should "most updated" really be "most up-to-date"?

-- L372. This subheading could be more informative by referring to the idea that human disturbance is assessed at a finer/smaller resolution/scale.

-- L429. Grammar: "our method's dissimilarity matrix."

-- L465. I think the debate is still open as to whether nestedness "increases the number of coexisting species by minimizing interspecific competition," especially when analysis is based on quantitative networks, see, for example, Staniczenko, P.P.A., Kopp, J.C. & Allesina, S. (2013). The ghost of nestedness in ecological networks. *Nature Communications*, 4, 1931.

-- L498. Explain "smooth terms" more.

-- L508. "1000 permutations" of what?

-- Fig. 5. Would be helpful to explain how negative and positive values on the y-axis should be interpreted.

Reviewer #3 (Remarks to the Author):

Strengths:

The paper is well-written and, on the whole, very clear. The scale of the study is relatively large – 196 networks. There are few high-profile papers at the global level showing how mutualistic interactions change over space and identifying potential drivers of these changes. The confirmation that network structure is consistent across ecoregions, biomes, and levels of human disturbance is notable. Finally, the authors went through extensive efforts to ensure high quality network data, as detailed in the supplemental materials. This is an admirable and well-documented effort that is sure to be of benefit to others, assuming the final networks are made publicly available, as planned (L517-8).

Major Concerns:

1. Overall, I find the manuscript unsurprising, largely because the results are logical extensions of established patterns (e.g. Smith et al 2018, Fricke et al 2020, Vizentin-Bugoni et al 2019). I think the work is solid and there was clearly a lot of effort put into data wrangling, but the main message that species interactions change across ecoregional and biome boundaries, and with levels of human disturbance is to be expected. There was a minor focus on the importance of these patterns for propagation of disturbances, which is an interesting outcome of the spatial patterns, but the analysis doesn't directly address this. The finding that network structure is similar across all regions confirms patterns found in a smaller number of networks by Vizentin-Bugoni et al 2019. Presentation of competing hypotheses for how ecological networks might change across space and environmental gradients/boundaries would make the manuscript more noteworthy.
2. I did not have enough information in L134-148 to understand the methodological approach, and it took a couple of reads through the methods to fully understand them. A figure that depicts the workflow (perhaps in the supplemental materials) would be helpful.
3. Even after reading through the methods, I still am not fully clear on how this analysis accounted for systematic biases associated with sampling methods (e.g. zoocentric vs phytocentric approaches will produce different networks). Are there spatial patterns in sampling methods that might influence results? In addition, the findings that interaction rewiring increases across various gradients is intriguing, but I was left wondering how much of the rewiring is simply due to inadequate or uneven sampling.
4. Human footprint is an important factor, and it is in the analysis twice, first as part of the environmental dissimilarity and then as a standalone factor. Does this unintentionally influence the results? In addition, sometimes it is referred to as footprint and other times as disturbance. It would be clearer to use the same phrase throughout.

Minor concerns:

L107: please provide some more detail on what types of interactions may be favored over others.

L189-195: Nice paragraph!

L206: why not all?

L270-277: What proportion of the world's land surface is covered by these biomes & ecoregions (and what proportion is not covered)? What proportion of land area is covered by tropical & subtropical broadleaf forests? Maybe one of the larger conclusions from this is that we haven't studied seed dispersal networks in most of the world's ecoregions?

L279-281: How does turnover vary between birds and plants?

L316-317: The #'s don't add up. $12 + 62 + 110 = 184$, but total is 196. Did you find the other 12 from the recently published study (Ref 10)? And how much overlap is there between this study and the papers in Ref 10?

L320: Add a year to indicate when the "most updated" map was published.

L515: Which predictors were correlated?

Figure 1: This is a useful figure, however it could use a few tweaks. The lines in A are light and hard to see, and the lines linking to New Zealand on the left side of the figure are confusing. And do the grey polygons represent ecoregions too?

Figure 2: Q-Q plots are hard to read. In addition, it was not immediately clear to me from the figure or legend what 'distance' means in "human footprint distance". I figured it out after reading the methods, but I suggest adding more details in order to make the figure interpretable on its own.

L526: SuppFig 17 – How can 67% be from distinct biomes if 70% are from the same biome?

REVIEWER COMMENTS

Reviewer #1 (Remarks to the Author):

In the present manuscript the authors study dissimilarity in observed avian frugivory networks and, in particular, the dependence on a variety of factors including biomes, ecotypes, elevation etc.

The authors raise interesting questions and probably invested much effort in assembling and standardizing the database. However from a data science perspective I find this manuscript rather weak. This type of data and is notoriously noisy and hence analysis must control for this. However, the paper just combines several fairly standard procedures and packages.

REPLY: We thank the reviewer for the opportunity to respond to these concerns. First, we acknowledge that most of the standardization procedures we used were previously explained in the Supplementary Material, where they were more likely to be missed. We have now made them clearer in the main manuscript (lines **158-168**), following a recommendation of Reviewer 2 and concerns raised by Reviewer 3.

In terms of the noise in the data, we carried out a standardization process for plant and bird taxonomy, and we used a greater number of sampling-related statistical controls (5 sampling-related metrics) than other papers in high profile journals (see some examples below). Specifically, we used several sampling-related metrics (hours, months, years, intensity and methods) to control for potential sampling artifacts in our analysis and potential ‘noise’ in our network data.

Doré, M., Fontaine, C. & Thébault, E. Relative effects of anthropogenic pressures, climate, and sampling design on the structure of pollination networks at the global scale. *Glob. Chang. Biol.* **27**, 1266–1280 (2021).

Schleuning, M. et al. Specialization of mutualistic interaction networks decreases toward tropical latitudes. *Curr. Biol.* **22**, 1925–1931 (2012).

From the reviewer’s comments, it is unclear if there are specific additional sources of noise which they feel should also be accounted for. From our perspective, we have accounted for as many as we could and in a manner consistent with or surpassing common practice. Finally, we highlight that our goal with this manuscript was to test several hypotheses rather than to develop new data science methods, and we are confident that the combination of existing methods that we used was sufficient to achieve this goal.

Essential bits such as the distance metrics are ad-hoc defined and don't take advantage of recent insights in network metrics. Analysis of complex relationships by a linear method is questionable and prone to lead to artifacts. The link between numerical results, hypothesis and conclusions is quite weak. The authors consider hypothesis based on structurally different factors (e.g. 1d continuous elevation vs. 2d discrete ecotype) and then evaluate with an approach that is agnostic of these structural differences, which again can easily yield misleading results. The results of the analysis seem plausible but this alone is insufficient to support the validity of the conclusions.

REPLY: We recognise that choosing metrics of network structure is not an easy task, as many network metrics are strongly correlated with one another and all authors and reviewers have their preferred metrics. Our decision was therefore to focus on a broad selection of metrics that represented different aspects of network structure, such as the number of links, their relative weightings, and their arrangement among species, and combine them into a single distance metric. The network metrics we used (weighted connectance, weighted nestedness, interaction evenness, PDI and modularity) are well-established in the literature and have been used in other large-scale studies of ecological networks (see some examples below).

Dugger, P. J. et al. Seed-dispersal networks are more specialized in the Neotropics than in the Afrotropics. *Glob. Ecol. Biogeogr.* **28**, 248–261 (2019).

Sebastián-González, E., Dalsgaard, B., Sandel, B. & Guimarães, P. R. Macroecological trends in nestedness and modularity of seed-dispersal networks: Human impact matters. *Glob. Ecol. Biogeogr.* **24**, 293–303 (2015).

Fricke, E. C. & Svenning, J. C. Accelerating homogenization of the global plant–frugivore meta-network. *Nature* **585**, 74–78 (2020).

Albouy, C. et al. The marine fish food web is globally connected. *Nat. Ecol. Evol.* **3**, 1153–1161 (2019).

We agree with the reviewer that the relationships between our predictors and response variables are complex and may not be captured by a linear method. However, we highlight that we used Generalized Additive Models (GAMs) specifically because of their flexibility in modelling non-linear relationships using smooth functions, which are represented by a sum of simpler, fixed basis functions that determine their complexity (please see lines **607-612**, where we added a description of why we used GAMs in our analyses).

We also highlight that all our variables (predictors and response) are distance-based 2D matrices, and our principal results were qualitatively the same when we used the binary versions of ecoregion and biome distance matrices (with two levels, ‘same’ or ‘distinct’) and their quantitative counterparts (a continuous variable based on the environmental dissimilarity between ecoregions/biomes) in our models. Thus, in addition to GAMs being able to handle different types of variables (i.e., categorical and continuous), our results were unchanged when we used only continuous variables in the models. Please note that we’ve added a new figure (**Figure 1**) that shows a summary of our methods, including how we tested the significance of our predictor variables by employing a combination of Generalized Additive Models and Multiple Regression on distance Matrices.

Finally, we have updated our hypotheses (please see lines **116-134** and **135-144**) and our Results section to better align these components. We believe that our updates have contributed to improve the link among hypotheses, numerical results, and conclusions.

In summary this the authors investigate an interesting question using an interesting dataset, but their methodology is not too simplistic to convincingly support the conclusions.

REPLY: We thank the reviewer for the comment and suggestions. We believe that the revised version of the manuscript provides a better description of our methods, and we

emphasise that employing complicated methodology was not our objective. On the contrary, for the broad readership of a journal such as *Nature Communications*, we believe that the simplest way to adequately test hypotheses is likely to be most accessible and thus achieves greatest impact.

Reviewer #2 (Remarks to the Author):

Review of "Global and regional ecological boundaries drive abrupt changes in avian frugivory interactions" by Martins et al.

The authors investigate how the composition and structure of avian frugivory networks varies within and between ecoregions and biomes, and also based on the level of human disturbance. They find that networks in different ecoregions and biomes contain dissimilar sets of species and interactions, with dissimilarity increasing with the level of human disturbance; however, network structure does not change significantly across ecoregions and biomes.

The authors address an interesting question using a great data set and the methods appear sound. However, the presentation needs to be improved: it is hard to follow what's going on regarding the actual analysis unless someone first reads the Methods section. Throughout the Introduction, Results and Discussion, there is too much unexplained jargon and insufficient explanation of methods such that I could not really understand what the authors did or the motivation for the study, nor interpret results. I expand on this below as three major comments: (i) Make the Introduction understandable to a general scientist and provide more description of methods in the main narrative, (ii) Include actual results in the main text, and (iii) Address some technical concerns. The authors have performed a lot of analyses, but it is by-and-large left up to the reader to piece together 23 figures (5 in main text plus 18 in SI) and 38 Tables (1 in main text plus 37 in SI) to understand what was done and what the results are. The work has the potential to be important and impactful, but this requires that the authors more clearly describe their main findings (in more straightforward language) and, more importantly, explain the supporting evidence.

REPLY: We thank the reviewer for the comments, and particularly for offering clear suggestions for improvement. We've updated all the sections of our manuscript to follow the reviewer's suggestions, and we agree that this has significantly improved readability. We describe below how we addressed each individual comment.

(i) Make the Introduction understandable to a general scientist and provide more description of methods in the main narrative

The Introduction, and especially the presentation of hypotheses, should be presented in a way that is understandable to a general scientist, or at least a typical ecologist. Right now, the audience appears to be limited to ecological network researchers.

The main hypothesis (L107), "that the large-scale distribution of species interactions is punctuated by ecoregion and biome boundaries," is too abstract. What exactly is being tested and why? Even the more specific hypothesis (L127)

contains jargon---"interaction dissimilarity," "changes in the whole structure of networks"---that is not defined (nor are the terms necessarily intuitive). What exactly is expected to change: the number and/or identity of interactions, species, value of network metrics?

REPLY: Following the reviewer's suggestion, we've updated our hypothesis, which now reads (lines **116-134**): *'Because species tend to be replaced across ecosystems^{2,4} and environmental conditions can favour some types of interactions over others (e.g., by altering the quality and detectability of interaction partners)¹⁵, we hypothesize that ecoregions and biomes delineate the large-scale distribution of species interactions. Specifically, we expect to find sharp differences in the composition of species interactions when crossing ecoregion and biome boundaries, beyond what would be expected from spatial processes alone – which are known to drive gradual changes in species and interaction composition¹⁵. Indeed, distance-decay relationships have been demonstrated across spatial and elevational gradients not only for species¹⁶, but also for ecological networks¹⁷⁻¹⁹, and likely result from dispersal limitation and increasing environmental dissimilarity with increasing geographic distance^{15,16}. Alternatively, ecological boundaries might be blurred by the processes of species and interaction homogenization (i.e., increasing similarity among biological communities), which accompany human disturbances such as land-use change and biotic invasions^{12,20}. Thus, an alternative hypothesis would be that shared interactions and biotic homogenization prevent any sharp discontinuities in interaction composition. If this is true, we expect to find a gradual decrease in the similarity of interactions with increasing spatial distance, but no abrupt differences in the identity of interactions from networks located at distinct ecoregions and biomes'.*

Finally, the more specific hypothesis now reads (lines **139-144**): *'Given known patterns of species turnover across environmental gradients¹⁶, we hypothesize a similar pattern of turnover in interaction composition (hereafter, interaction dissimilarity), which could potentially lead to changes in the whole structure of networks (i.e., changes in the arrangement of interactions among species), represented here by a metric combining several descriptors of network architecture, which we call 'network structural dissimilarity' (see Methods for more details)'.*

The authors assume the reader knows what an avian frugivory network is (L124), whereas it would be safer to explain what the nodes and links represent. Similarly, they should explain what weights represent in their quantitative networks (L135).

REPLY: We agree with the reviewer and added the following sentences to describe what an avian frugivory network is and what are the weights in quantitative networks:

*'We focused on avian frugivory networks, that is, local communities of interacting bird and fruiting plant species, because of their importance for seed dispersal²¹, promoting species diversity²² and regenerating degraded ecosystems²³' (lines **144-147**) and 'Local networks are composed of nodes – plant and bird species, connected by links whenever two species interact with each other. Each local network is represented by a matrix, with plants and birds on rows and columns, respectively, and cell values describing the weighted network links – the number of fruit-feeding events (i.e., interaction frequency) between a plant and bird species.' (lines **153-158**).*

There also needs to be more description of methods in the main narrative, so that the reader doesn't have to constantly jump back-and-forth to the Methods section (without adequate direction, it must be noted) or SI. Indeed, the first part of only two descriptions of methods in the main narrative invites the reader to look at 6 supplementary figures and 2 supplementary tables to understand how they accounted for sampling effects (L137): "To ensure that our results would not be driven by taxonomic uncertainty and sampling effects, we standardized the taxonomy of plant and bird species in our networks following a series of steps (Supplementary Figs. 3-6) and controlled statistically for network sampling metrics in our analyses (Supplementary Figs. 7-8; Supplementary Tables 2-3)." Even a few sentences describing what was actually done would be helpful. Then the second part describing methods is, I feel, insufficient for the reader to form even the vaguest intuition of what analysis was performed (L143): "We generated several distance matrices ($N \times N$, where N is the number of local networks in our dataset) to be our variables in the statistical models. Specifically, we used ecoregion, biome, local human disturbance (measured using the human footprint index), spatial, elevation and sampling-related (i.e., hours, months, years, intensity and methods) distance matrices as predictor variables, and facets of network dissimilarity (i.e., species turnover, interaction dissimilarity, and network structural dissimilarity) as response variables."

REPLY: We agree, and following the reviewer's suggestion, we've updated the description of our methods at the end of the Introduction (please see lines 158-193). We've also added a new figure (**Figure 1**) that shows a summary of our methods, including how we generated the distance matrices used in our models.

Moreover, I don't think the authors satisfactorily explain why studying "network dissimilarity" (their response variable) is interesting in the first place.

REPLY: We've added new sentences to describe the importance of evaluating different facets of network dissimilarity (lines 175-181): *'By evaluating these three different facets of network dissimilarity, we were able to assess the extent to which changes in species composition are associated with changes in both the identity of component interactions (interaction dissimilarity) and the architecture of local networks (network structural dissimilarity, which may remain the same despite turnover of species and interactions^{25,26}). Together these facets contribute to greater understanding of the scale at which one ecological network ends and another begins, and how/why networks vary across large spatial scales^{15,25}'.*

(ii) **Include actual results in the main text**

There should be more numerical results in the main narrative, along with more informative summaries of observed trends. For example, the opening of the Results section simply tells the reader to go look at 3 figures and 2 tables in the SI to get an idea of the actual results (L152): "We found that the turnover of plant and frugivorous-bird species composition was strongly affected by ecoregion and biome boundaries (Supplementary Tables 4 and 5; Supplementary Figs. 9a-b). Similarly, there was an overall trend of networks located at different positions along the human disturbance gradient having different species composition (Supplementary

Fig. 9c)." A large part of the issue is that the reader encounters the results without really understanding what analysis was done.

Even 4 of the 5 figures in the main text are covered in only 3 sentences (L165):

"This provides strong support to the hypothesis that large-scale ecological boundaries mark spatially abrupt changes in plant-frugivore interactions (Figs. 1 and 2; Supplementary Fig. 11). Importantly, a great proportion of the deviance explained by biomes was shared with ecoregions (Fig. 3; Supplementary Fig. 12), which suggests that changes in interaction dissimilarity across biome boundaries mostly reflect the variation occurring at a finer (ecoregion) scale. In fact, crossing an ecoregion boundary induced an average 7% increase in interaction dissimilarity, while crossing a biome boundary induced only an additional 0.2% change. As with species, networks located at opposite ends of the human disturbance continuum usually exhibited very different interactions (Fig. 4; Supplementary Fig. 13)." Again, at this point in the text, it wasn't clear how to interpret these results, given the so far limited explanation of methods.

Another example is the additional description of Figure 3 (L189): "The shared effect of ecoregion boundaries and spatial distance explained the greatest proportion of the variability in plant-frugivore interaction dissimilarity, followed by the unique contributions of these two variables (Fig. 3)." It is not clear what "shared effect" means, nor "proportion of the variability in plant-frugivore interaction dissimilarity," nor "unique contributions to these two variables" ---how is a boundary expressed as a variable?

REPLY: We agree with the reviewer and have updated our entire Results section to include more numerical results and descriptions of the reported patterns. We've also added a description of the deviance partitioning analyses to the end of the Introduction (lines **189-193**), allowing the reader to understand what unique and shared effects mean before getting to the Results section.

There are some very interesting observations in the Discussion section (L215 to L221), but supporting evidence needs to be presented, explained, and linked in the main narrative. I would also like to see more discussion on (L248), "these results suggest that disturbances in local networks of frugivory are much less likely to impact networks from distant sites or elevations, especially if the networks are located within distinct ecoregions and biomes;" specifically, regarding the mechanisms that can lead to long-distance effects, i.e., how does one go from: (i) there are two networks separated by a large distance, (ii) one network experiences a disturbance, and (iii) this affects the second network? And (L285), "suggesting that the ecological processes underlying the architecture of frugivory networks may be independent of species and interaction composition" ---which processes?

REPLY: Following the reviewer's suggestions, we've updated our Discussion section. More specifically, we now discuss the mechanisms behind the propagation of disturbances across space in lines **323-331** and exemplify some processes underlying the architecture of networks (e.g., ecological specialization and species' functional roles) that may be independent of species and interaction composition (please see lines **380-382**). Lastly, following previous comments about the Introduction/hypothesis, we

hope that the revised introduction (specifically lines **100-115**) will better prepare the reader for how the reviewer's points **i**, **ii** and **iii** are connected.

(iii) Address some technical concerns

The authors remove non-avian frugivores from analyses (L297)---how much does this affect results?

REPLY: We have now included the following paragraph in the Methods section to explain in more detail our decision of removing non-avian frugivores from our analyses (lines **393-402**): *‘Importantly, we removed non-avian frugivores from our analyses because only 28 out of 196 raw networks (before data cleaning) sampled non-avian frugivores, and not removing non-avian frugivores would generate spurious apparent turnover between networks that did vs. did not sample those taxa. In addition, the removal of non-avian frugivores did not strongly decrease the number of frugivores in our dataset (Supplementary Fig. 20a) or the total number of links in the global network of frugivory (Supplementary Fig. 20b). Furthermore, non-avian frugivores, as well as their interactions, were not shared across ecoregions and biomes (Supplementary Fig. 21), so their inclusion would only strengthen the results we found (though as noted above, we believe that this would be spurious because they are not as well sampled)’.*

Finally, we highlight that because networks are a snapshot of local communities of interacting species, delimited by factors such as sampling area and focal taxa, it is plausible to assume that most networks that sampled only birds were located in sites where other taxa (e.g., mammals, reptiles, fishes) consume fruits. By removing these additional taxa from our analyses, we are mirroring a process that was employed in the field by all researchers who only sampled birds (either by defining a sampling method that only captures bird interactions, such as mist netting, or by defining birds as the only focal taxon).

The minimum size of network analyzed can be very small (potentially containing only 4 species, L299). It would be informative to see some data/network summary statistics in the main text. I worry that much of the analysis, especially of network properties, is not suitable for such small networks; I would recommend setting a minimum network size to something more like 8 to 12 species---how many networks would that rule out from analysis?

REPLY: We thank the reviewer for the comment. We agree that changing the threshold for minimum network size has the potential to affect the reported patterns, especially those regarding network structural dissimilarity. We highlight, however, that although our criterion established a minimum of three species for each trophic level (we only included networks with greater than two species in each trophic level), none of our local networks had only six species in total. In fact, the smallest size of a local network used in our analyses was eight species, but this represented only six local networks in our dataset (~3% of our local networks; please see Supplementary Fig. 22a).

Nevertheless, this size threshold for including networks is quite arbitrary, so rather than picking another higher (but equally arbitrary) threshold, we now perform a sensitivity analysis to evaluate how sequentially removing all networks below a specified threshold of size (i.e., classes of network sizes) would change the estimates (t and F values) and

significance of our predictor variables. We did this by removing all classes of network sizes in our dataset from smallest to largest, such that in the final round of the sensitivity analysis all classes of network size (i.e., all local networks) would be removed (Supplementary Fig. 22b). Note, however, that although we had 60 different classes of network size (minimum value = 8 species; maximum value = 238 species), we could perform the analysis only up to the removal of all networks with 71 (or fewer) species (which represented 183 out of the 196 local networks in our dataset). We could not remove networks with larger sizes because further removing networks would cause the GAMs to have more coefficients than data (i.e., the analyses would have too few data points).

Importantly, we found that removing small networks from our dataset did not strongly affect our results; for instance, removing networks with up to 10 species (which represent 17 out of 196 local networks in our dataset) would not affect any of the reported patterns (please see Supplementary Figs. 23 and 24). Moreover, almost all the significant effects in our full models were still significant even after the removal of networks with up to 20 species (which represented ~ 42% of the local networks in the dataset). The only exception was the effect of biome boundaries on interaction dissimilarity, which seems to be more sensitive to the sequential removal of small networks (Supplementary Fig. 23). Note, however, that biome boundaries explained a relatively low unique proportion of the variation in interaction dissimilarity in our full model anyway, as most of the deviance explained by biomes was shared with ecoregions (Supplementary Fig. 12), which is likely because biomes share boundaries with ecoregions and the latter explain finer-resolution environmental differences.

We also highlight that even though all estimates tend to approach zero with the removal of larger networks, this is partially because beyond a certain number of network removals there are too few data points and insufficient range of the predictor value for the model to be able to detect an effect. Please see lines **302-338** of the Supplementary Information for more details on this sensitivity analysis.

Minor comments

-- L204. Explain, "all our main results were robust to different processes of assigning uniqueness to problematic species (i.e., species without a valid epithet) (Supplementary Tables 9-33)."

REPLY: We've added a sentence in lines **274-276** to describe in more detail the process of assigning uniqueness to problematic species.

-- L241. Explain, "at least one region where novel interactions performed by introduced species have largely replaced those performed by native species..."

REPLY: We've updated this sentence (please see lines **313-319**) to describe how the decline and extinction of native species could lead to large-scale connections in the global network of frugivory.

-- L320. Should "most updated" really be "most up-to-date"?

REPLY: Following the reviewer's suggestion, we've replaced 'most updated' to 'most up-to-date'.

-- **L372. This subheading could be more informative by referring to the idea that human disturbance is assessed at a finer/smaller resolution/scale.**

REPLY: We thank the reviewer for the comment and replaced the former subheading by 'Local-scale human disturbance distance'.

-- **L429. Grammar: "our method's dissimilarity matrix."**

REPLY: We've updated the grammar.

-- **L465. I think the debate is still open as to whether nestedness "increases the number of coexisting species by minimizing interspecific competition," especially when analysis is based on quantitative networks, see, for example, Staniczenko, P.P.A., Kopp, J.C. & Allesina, S. (2013). The ghost of nestedness in ecological networks. Nature Communications, 4, 1931.**

REPLY: We agree with the reviewer and removed this sentence from the Methods section.

-- **L498. Explain "smooth terms" more.**

REPLY: We have added a new sentence to describe in more detail what are smooth functions. Because smooth terms are the smooth functions defined for each predictor in the model, we believe that this will help readers to better understand how our GAM-based MRM models work. Please see lines **607-612**.

-- **L508. "1000 permutations" of what?**

REPLY: Here, we are referring to the permutation (shuffling) of the response distance matrix (i.e., facets of network dissimilarity). We now explain this in lines **623-624**.

-- **Fig. 5. Would be helpful to explain how negative and positive values on the y-axis should be interpreted.**

We've added the sentence '*Partial residuals remain on the same scale as the original data, but the sign of values indicates how they differ from what would be expected (i.e., from the fitted values) based on the other predictor variables in the model*' in the legend of the figure (which is now **Figure 6**).

Reviewer #3 (Remarks to the Author):

Strengths:

The paper is well-written and, on the whole, very clear. The scale of the study is relatively large – 196 networks. There are few high-profile papers at the global level showing how mutualistic interactions change over space and identifying

potential drivers of these changes. The confirmation that network structure is consistent across ecoregions, biomes, and levels of human disturbance is notable. Finally, the authors went through extensive efforts to ensure high quality network data, as detailed in the supplemental materials. This is an admirable and well-documented effort that is sure to be of benefit to others, assuming the final networks are made publicly available, as planned (L517-8).

REPLY: We thank the reviewer for the positive comments.

Major Concerns:

Overall, I find the manuscript unsurprising, largely because the results are logical extensions of established patterns (e.g. Smith et al 2018, Fricke et al 2020, Vizentin-Bugoni et al 2019). I think the work is solid and there was clearly a lot of effort put into data wrangling, but the main message that species interactions change across ecoregional and biome boundaries, and with levels of human disturbance is to be expected.

REPLY: We agree with the reviewer that our results are logical extensions of previously demonstrated patterns. Indeed, we were reassured to see that our results aligned with our broader expectations. However, we highlight that the field of biogeography of interactions is still in its infancy, and no study has previously evaluated the matching of species and interaction patterns across large-scale ecological boundaries. Furthermore, although the association between species and interaction composition is a logical hypothesis, interaction rewiring has the potential to decouple this relationship, and we showed significant rewiring of interactions across distance and disturbance gradients. In other words, in these cases interactions responded more than you would expect for the changes in species composition. This can be seen as a logical extension of recent findings (and it is indeed logical) but it's one that paves the road for a deeper understanding of the biogeography of biotic interactions. Moreover, our findings that interaction composition, but not network structure, changes across ecoregions and biomes suggest that structural differences observed previously across habitats do not appear to have a biogeographic-scale analog.

There was a minor focus on the importance of these patterns for propagation of disturbances, which is an interesting outcome of the spatial patterns, but the analysis doesn't directly address this. The finding that network structure is similar across all regions confirms patterns found in a smaller number of networks by Vizentin-Bugoni et al 2019.

REPLY: We agree with the reviewer that discussing the propagation of disturbances across the global network of frugivory is an interesting outcome of our findings. We now discuss this in lines **323-334** of the 'Discussion' section. Our results that network structure does not change across ecological boundaries do indeed confirm previous findings by Vizentin-Bugoni et al 2019 at a smaller scale, and we've included this reference in our Discussion section.

Presentation of competing hypotheses for how ecological networks might change across space and environmental gradients/boundaries would make the manuscript more noteworthy.

REPLY: Following the reviewer's suggestion, we've included two alternative hypotheses in the Introduction section. Please see lines **116-126** (Main hypothesis) and lines **126-134** (Alternative hypothesis).

I did not have enough information in L134-148 to understand the methodological approach, and it took a couple of reads through the methods to fully understand them. A figure that depicts the workflow (perhaps in the supplemental materials) would be helpful.

REPLY: We thank the reviewer for the suggestion. We've now added a description of the methods used in the end of the Introduction section (please see lines **158-193**) and a new figure that depicts our workflow, including how we generated our distance matrices and used them in our GAM-based MRM models (please see what is now **Figure 1**).

Even after reading through the methods, I still am not fully clear on how this analysis accounted for systematic biases associated with sampling methods (e.g. zoocentric vs phytocentric approaches will produce different networks).

REPLY: Our variable named 'sampling focus', which was one of the variables used to generate our 'sampling methods' distance matrix, captures whether a zoocentric or phytocentric approach was used (i.e., whether the focal organisms were birds, plants, or both). We agree with the reviewer that the previous description wasn't clear enough, so we've added the following sentence in the description of this variable: '*As such, this variable determines if authors used a zoocentric or a phytocentric sampling method (or a combination of the two)*' in Supplementary Table 3. We've also added another sentence in the Methods section (line **535-540**), which now reads: '*Because sampling methods, such as sampling design, focus (i.e., focal taxa, which determines whether a zoocentric or phytocentric method was used), interaction frequency type (i.e., how interaction frequency was measured) and coverage (total or partial) might also affect the observed plant-frugivore interactions⁴¹, we combined these variables into a single distance matrix to estimate the overall differences in sampling methods between networks*'.

Are there spatial patterns in sampling methods that might influence results?

REPLY: We thank the reviewer for raising this issue. We've added a new figure in our Supplementary Information (Supplementary Fig. 19) demonstrating that the correlations between our sampling-related metrics and spatial distance are not strong (all Pearson's correlation coefficients ≤ 0.3) (please note that our deviance partitioning analyses already showed the unique and shared effects of the predictors that were not used for controlling sampling artifacts). Furthermore, we highlight that we tested our hypotheses using test statistics (t and F values) from the model (GAM) partial coefficients. As such, we're testing the independent effect of each predictor, holding any other correlated predictors (including sampling methods) constant.

In addition, the findings that interaction rewiring increases across various gradients is intriguing, but I was left wondering how much of the rewiring is simply due to inadequate or uneven sampling.

REPLY: We agree with the reviewer that sampling differences could promote interaction rewiring. Because we've included several sampling-related metrics in our analyses, we are confident that we have done the best job possible to account for the effect of sampling artifacts in our response variables. Indeed, differences in sampling effort (e.g., one network is well-sampled, while the other is not) should be captured by our sampling-related variables (sampling hours, months, years, methods, and intensity). This is corroborated by the fact that we've found that differences in sampling hours, intensity and methods were strong drivers of interaction rewiring (please see Supplementary Table 36). However, as noted in the previous response above, our testing of partial coefficients tests for independent effects of each predictor, holding constant the effects of all others (including sampling intensity measures) in the model. As such, sampling effort may add to, but not confound, the results presented.

Human footprint is an important factor, and it is in the analysis twice, first as part of the environmental dissimilarity and then as a standalone factor. Does this unintentionally influence the results? In addition, sometimes it is referred to as footprint and other times as disturbance. It would be clearer to use the same phrase throughout.

REPLY: We thank the reviewer for the opportunity to clarify this. In the main analysis presented in the manuscript, human footprint is only used once (as our measure of disturbance). In addition, we repeated the main analysis but using quantitative versions of ecoregion and biome distance matrices (presented in the Supplementary Material), which accounted for the amount of environmental dissimilarity between ecoregions/biomes. Although it is true that ecoregion and biome-scale averages of human footprint were one of the variables used to calculate these quantitative versions of ecoregion and biome distances in the supplementary analysis, local-scale human footprint had a strong partial effect on our response variables when either the binary (main text) or quantitative (supplement) versions of ecoregion and biome distance matrices were used in the models. Moreover, we highlight that local-scale human footprint was measured within a 5-km buffer zone around each network site, while ecoregion and biome-scale footprint were averaged at very large scales (over the entire ecoregion or biome), so could differ considerably from the local values.

We agree with the reviewer that using both terms (footprint and disturbance) can be confusing for the reader. Hence, we've tried to make clear that human footprint was used to estimate human disturbance in **Figure 1**. Furthermore, we've replaced the term human footprint by human disturbance throughout the text.

Minor concerns:

L107: please provide some more detail on what types of interactions may be favored over others.

REPLY: Following the reviewer's suggestion, we've added the sentence '(e.g., by altering the quality and detectability of interaction partners)' in lines **117-118**.

L189-195: Nice paragraph!

REPLY: We thank the reviewer for the comment.

L206: why not all?

REPLY: Although most of the reported patterns remain qualitatively the same when removing individual studies from the dataset, some of the patterns we observed were more strongly driven by individual studies with multiple networks and are therefore less representative. For example, the effect of biome boundaries on interaction dissimilarity is no longer significant when we remove the study with greatest number of networks ($N = 35$) in our dataset from the analyses. Furthermore, some of the variables used to control for sampling artifacts had variable effects when removing individual studies (potentially because few studies used each specific combination of sampling methods). Importantly however, the effect of ecoregion boundaries, human disturbance, spatial and elevation gradients remained unchanged even with the removal of individual studies. We explain this in more detail in the Supplementary Information (lines 254-274).

L270-277: What proportion of the world's land surface is covered by these biomes & ecoregions (and what proportion is not covered)? What proportion of land area is covered by tropical & subtropical broadleaf forests? Maybe one of the larger conclusions from this is that we haven't studied seed dispersal networks in most of the world's ecoregions?

REPLY: Following the reviewer's comment, we've added the sentence '*We also highlight that the ecoregions and biomes represented in our dataset cover around 20% and 69%, respectively, of the world's ice-free land surface. As such, network sampling in data deficient regions³⁵, especially at the ecoregion scale, may contribute greatly to our understanding of macroecological patterns in avian frugivory networks*' in lines 366-370. We also updated the legend of our Supplementary Figure 2 to '*Geographic distribution of the 196 avian frugivory networks in our dataset. Local networks were distributed across 11 biomes, with most of these being located within a single biome, the Tropical & Subtropical Moist Broadleaf Forests, which covers around 11% of the world's ice-free land surface*' (please see lines 363-366) of the Supplementary Information).

L279-281: How does turnover vary between birds and plants?

REPLY: Following the reviewer's suggestion, we've added the sentence: '*This is corroborated here by the fact that networks located in distinct ecoregions and biomes tended to share more bird than plant species (Supplementary Fig. 17)*' in lines 375-376. Please also see Supplementary Fig. 17, where we show the connections between networks sharing bird and plant species, as well as pie charts depicting the proportion of pairs of local networks sharing bird/plant species in our dataset.

L316-317: The #'s don't add up. $12 + 62 + 110 = 184$, but total is 196. Did you find the other 12 from the recently published study (Ref 10)? And how much overlap is there between this study and the papers in Ref 10?

REPLY: We obtained 12 networks by checking the database available in Fricke and Svenning 2020. Importantly, although several networks present in our dataset are also present in Fricke and Svenning 2020, we highlight that we've used different criteria for selecting networks. For example, while we only collated quantitative networks including avian frugivores, Fricke and Svenning's dataset also includes binary networks involving other taxa (mainly mammals). We explain in more detail the reasons why we only selected quantitative networks in lines **590-593** and justify why we opted for removing non-avian frugivores from our analyses in lines **393-402**. Moreover, aiming to mitigate potential sampling artifacts, we only included networks in which sampling was not taxonomically restricted below birds or plants (to avoid overestimating network dissimilarity because of *e.g.*, differences in the bird family sampled).

L320: Add a year to indicate when the “most updated” map was published.

REPLY: Following the reviewer's suggestion, we've added this information in line 427.

L515: Which predictors were correlated?

REPLY: We've added a new figure showing the correlations between our predictors (please see Supplementary Fig. 19).

Figure 1: This is a useful figure, however it could use a few tweaks. The lines in A are light and hard to see, and the lines linking to New Zealand on the left side of the figure are confusing. And do the grey polygons represent ecoregions too?

REPLY: We thank the reviewer for the comments. We've updated our figure (now **Figure 2**) to make lines easier to visualize and added a sentence in the legend (lines **859-860**) explaining that lines disappearing at the side edges of the world map are connected to those from the opposite edge.

Please also note that lines were plotted along the great circle distance between network sites. Because our map is two dimensional, paths may not seem like the shortest distance between two networks (even though they represent the shortest distance between two networks on the surface of the globe). Finally, we highlight that all coloured areas (including the grey polygons) represent ecoregions where networks were located.

Figure 2: Q-Q plots are hard to read. In addition, it was not immediately clear to me from the figure or legend what 'distance' means in “human footprint distance”. I figured it out after reading the methods, but I suggest adding more details in order to make the figure interpretable on its own.

REPLY: We thank the reviewer for the suggestions. We've made the proposed changes in Figures 3 and 4.

L526: SuppFig 17 – How can 67% be from distinct biomes if 70% are from the same biome?

REPLY: Around 67% of the long-distance network comparisons (i.e., 67% of the pairs of networks that are located > 10,000 km of distance from each other) involve networks that belong to distinct biomes. On the other hand, 70% of the long-distance network connections (i.e., 70% of the pairs of networks that are located > 10,000 km from each other and share interactions), involve networks that belong to the same biome. We've updated the figure (which is now **Supplementary Fig. 18**) and its legend (please see lines **571-576** in the Supplementary Information) to make this clearer to the reader.

REVIEWERS' COMMENTS

Reviewer #2 (Remarks to the Author):

Review of revised "Global and regional ecological boundaries drive abrupt changes in avian frugivory interactions" by Martins et al.

I am Reviewer 2 from the first round of reviews. I have read the revised manuscript and the authors' response and I am satisfied with their replies and changes to the text. I am supportive of publication.

Reviewer #3 (Remarks to the Author):

The authors have thoughtfully responded to reviewer suggestions, and the manuscript is now much easier to follow and more accessible to a broad audience. In particular, I like Figure 1, as it clearly depicts the analytical approach. I am satisfied with nearly all of the responses and changes made, with the minor exception noted below.

The introduction now nicely introduces hypotheses for species composition and interaction composition, but there is still little development for how network structure might (or might not) change across ecoregions or biomes (L141-142 only briefly touches on this). References to existing literature could justify the inclusion of network structure in this study - here are a couple of papers that may be useful:

Dugger, P.J., Blendinger, P.G., Böhning-Gaese, K., Chama, L., Correia, M., Dehling, D.M., et al. (2018). Seed-dispersal networks are more specialized in the Neotropics than in the Afrotropics. *Global Ecol Biogeography*.

Sebastián-González, E., Dalsgaard, B., Sandel, B. & Guimarães Jr, P.R. (2015). Macroecological trends in nestedness and modularity of seed-dispersal networks: human impact matters. *Global Ecol Biogeography*, 24, 293–303.

REVIEWERS' COMMENTS

Reviewer #2 (Remarks to the Author):

Review of revised "Global and regional ecological boundaries drive abrupt changes in avian frugivory interactions" by Martins et al.

I am Reviewer 2 from the first round of reviews. I have read the revised manuscript and the authors' response and I am satisfied with their replies and changes to the text. I am supportive of publication.

We thank the reviewer for the comment.

Reviewer #3 (Remarks to the Author):

The authors have thoughtfully responded to reviewer suggestions, and the manuscript is now much easier to follow and more accessible to a broad audience. In particular, I like Figure 1, as it clearly depicts the analytical approach. I am satisfied with nearly all of the responses and changes made, with the minor exception noted below.

The introduction now nicely introduces hypotheses for species composition and interaction composition, but there is still little development for how network structure might (or might not) change across ecoregions or biomes (L141-142 only briefly touches on this). References to existing literature could justify the inclusion of network structure in this study - here are a couple of papers that may be useful:

Dugger, P.J., Blendinger, P.G., Böhning-Gaese, K., Chama, L., Correia, M., Dehling, D.M., et al. (2018). Seed-dispersal networks are more specialized in the Neotropics than in the Afrotropics. *Global Ecol Biogeography*.

Sebastián-González, E., Dalsgaard, B., Sandel, B. & Guimarães Jr, P.R. (2015).

Macroecological trends in nestedness and modularity of seed-dispersal networks: human impact matters. *Global Ecol Biogeography*, 24, 293–303.

We thank the reviewer for the comment. We agree that we could focus more on how network structure might change across ecological boundaries, so we added one of the suggested references (Dugger et al. 2018) and the following sentence in the Introduction section (lines 147-150):

'Notably, environmental conditions may also affect niche partitioning and interaction specialization, potentially explaining further structural differences among ecological networks from distinct habitats and biogeographical regions^{15,21,22}.'